# Adaptation to chronic ER stress enforces pancreatic β-cell plasticity

Chien-Wen Chen [1] ✉, Bo-Jhih Guan[1], Mohammed R. Alzahrani[2], Zhaofeng Gao[1], Long Gao[3], Syrena Bracey[1], Jing Wu[1], Cheikh A. Mbow[1], Raul Jobava[1], Leena Haataja [4], Ajay H. Zalavadia [5], Ashleigh E. Schaffer [1], Hugo Lee[6], Thomas LaFramboise[1], Ilya Bederman[1], Peter Arvan [4], Clayton E. Mathews [7], Ivan C. Gerling[8], Klaus H. Kaestner [3], Boaz Tirosh [2,9], Feyza Engin [6,10] ✉ & Maria Hatzoglou [1] ✉

Pancreatic β-cells are prone to endoplasmic reticulum (ER) stress due to their role in insulin secretion. They require sustainable and efficient adaptive stress responses to cope with this stress. Whether episodes of chronic stress directly compromise β-cell identity is unknown. We show here under reversible, chronic stress conditions β-cells undergo transcriptional and translational reprogramming associated with impaired expression of regulators of β-cell function and identity. Upon recovery from stress, β-cells regain their identity and function, indicating a high degree of adaptive plasticity. Remarkably, while β-cells show resilience to episodic ER stress, when episodes exceed a threshold, β-cell identity is gradually lost. Single cell RNA-sequencing analysis of islets from type 1 diabetes patients indicates severe deregulation of the chronic stress-adaptation program and reveals novel biomarkers of diabetes progression. Our results suggest β-cell adaptive exhaustion contributes to diabetes pathogenesis.

Pancreatic β-cells have an active secretory pathway, and may reach a production rate as high as 1 million molecules of preproinsulin per minute under stimulated conditions[1]. Therefore, tolerance to ER stress is an integral component of the normal physiology of β-cells[2]. In addition to physiological stress, several pathological conditions can elicit severe ER stress in β-cells, including reactive oxygen species, toxins, viral infections, and inflammation[3]. These conditions trigger the unfolded protein response (UPR), which is mediated by the coordinated actions of ER membrane-localized inositol-requiring protein 1 alpha (IRE1α), activating transcription factor 6 (ATF6) and the eIF2α PKR-like endoplasmic reticulum kinase (PERK)[4]. Under acute ER stress, the adaptive UPR establishes a return to cellular homeostasis, while chronic and unresolvable stress can result in a maladaptive UPR that contributes to pathology, including diabetes[5,6]. However, there is a time frame during which β-cells induce adaptive mechanisms[7].

While IRE1α activity has been implicated in the transition from adaptive to maladaptive UPR[8,9], the impact of recovery from chronic stress on β-cell identity is not well-defined. Several lines of evidence

[1]Department of Genetics and Genome Sciences, Case Western Reserve University, Cleveland, OH 44106, USA. [2]Department of Biochemistry, Case Western Reserve University, Cleveland, OH 44106, USA. [3]Department of Genetics and Institute for Diabetes, Obesity and Metabolism, University of Pennsylvania Perelman School of Medicine, Philadelphia, PA 19104, USA. [4]The Division of Metabolism, Endocrinology & Diabetes, University of Michigan Medical Center, Ann Arbor, MI 48105, USA. [5]Lerner Research Institute, Cleveland Clinic, 9620 Carnegie Ave N Bldg, Cleveland, OH 44106, US. [6]Department of Biomolecular Chemistry, University of Wisconsin-Madison, School of Medicine and Public Health, Madison, WI 53706, USA. [7]Department of Pathology, Immunology and Laboratory Medicine, University of Florida College of Medicine, Gainesville, FL, US. [8]Department of Medicine, University of Tennessee, Memphis, TN, US. [9]The Institute for Drug Research, The Hebrew University of Jerusalem, Jerusalem, Israel. [10]Department of Medicine, Division of Endocrinology, Diabetes & Metabolism, University of Wisconsin-Madison, School of Medicine and Public Health, Madison, WI 53705, USA. ✉e-mail: cxc1036@case.edu; fengin@wisc.edu; mxh8@case.edu

implicate ER stress and maladaptive UPR in the pathogenesis of type 1 and type 2 diabetes (T1D and T2D)[9–18]. Indeed, mutations in genes that influence ER stress responses, including those encoding the major UPR elements PERK, eIF2α, CHOP, XBP1, IRE1α, and WSF1, frequently result in loss of β-cell function and/or death in experimental models and humans[3,19–21]. In this study, we investigated the molecular underpinnings of reversible chronic and episodic ER stress and their effects on β-cell adaptation.

The use of chemical stressors is critical to observe temporal changes during the progression from acute to chronic ER stress in cells[22] and in vivo[23–26]. Genetic chronic ER stress models[27–29] do not provide the necessary synchronized control over these transitions; however, numerous studies show that the biochemical principles of

the ER stress response are shared between chemically-induced stress in vitro and in vivo[24–26]. Therefore, we used the reversible inhibitor of the sarcoendoplasmic reticulum pump $Ca^{2+}$ ATPase (SERCA) pump, cyclopiazonic acid (CPA), to model resolvable chronic ER stress in the mouse insulinoma cell line, MIN6. This inhibitor induces ER stress that can be tuned in duration by washing out[22]. Using CPA, we showed that β-cells undergo a state change associated with adaptive reprogramming of their transcriptome and translatome that compromises β-cell identity. A comparative analysis of adaptive reprogramming between β-cells (MIN6) and mouse embryonic fibroblasts (MEFs) revealed an elaborate, β-cell-specific network of genes that exhibit stress-induced expression changes. Adaptation to chronic ER stress in β-cells included changes in expression of 334 genes, with a predominant

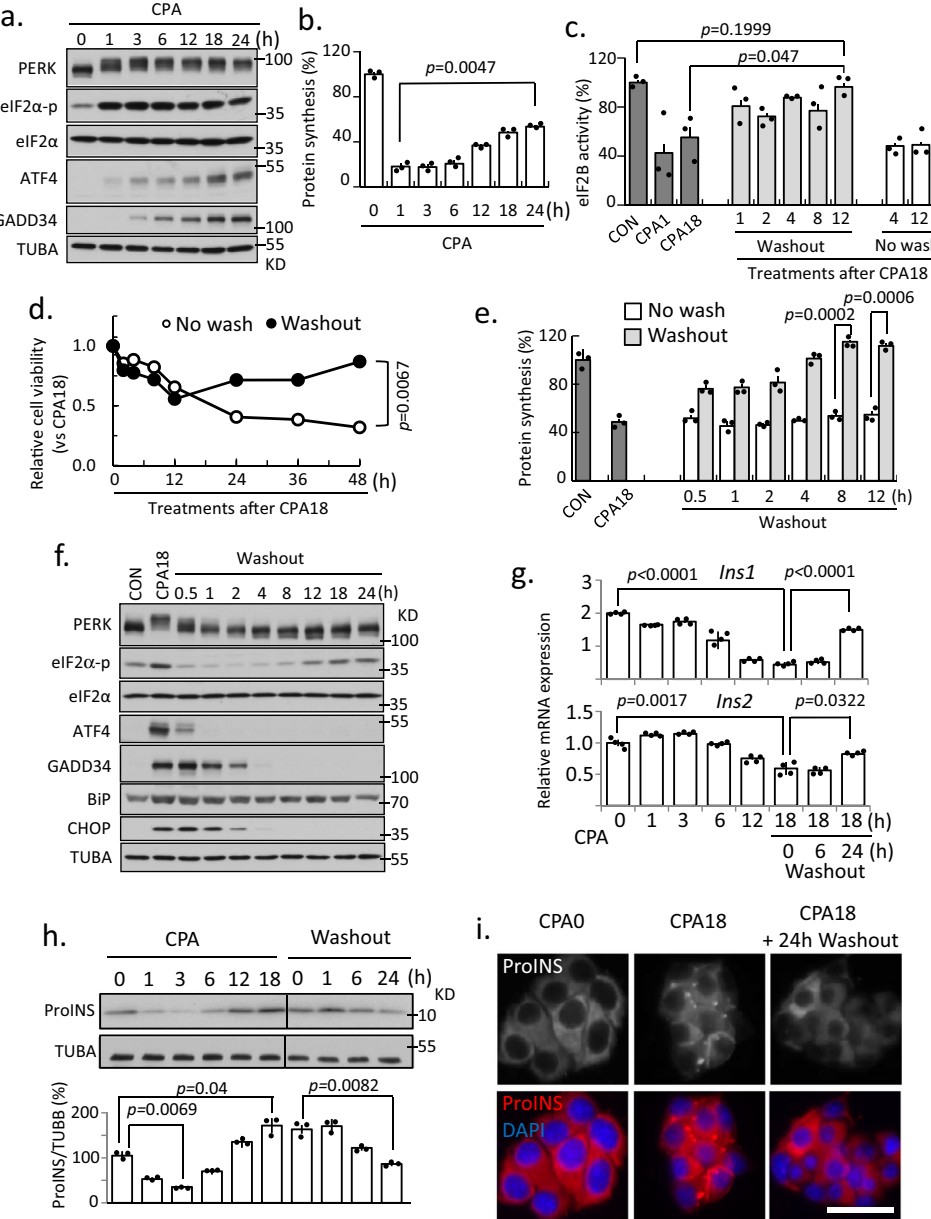

**Fig. 1 | MIN6 cells can simultaneously survive chronic ER stress and alter proinsulin proteostasis.** Western blot analysis (**a**), protein synthesis, $n = 3$ (**b**), and eIF2B GEF activity, $n = 3$ (**c**), measured in MIN6 cells treated with CPA as indicated. Cell viability (**d**), protein synthesis, $n = 3$ (**e**), and Western blot analysis (**f**) measured in MIN6 cells for the indicated treatments. qRT-PCR analysis for *Ins1 and Ins2* mRNA levels normalized to *GAPDH*, $n = 3$ (**g**), and Western blot analysis for proinsulin, $n = 3$ (**h**), measured in MIN6 cells for the indicated treatments. **i** Fluorescence

immunocytochemistry of proinsulin at 0 and 18 h of CPA treatment and 24 h of washout following CPA treatment. Scale bar represents 50 µm. Two independent experiments; The representative images were shown. Error bars represent S.E.M. *p*-value represents the statistical test by two-tailed paired Student's *t*-test. Representative western blotting images were shown. Dots in all plots represent independent experiments. Source data are provided as a Source Data file.

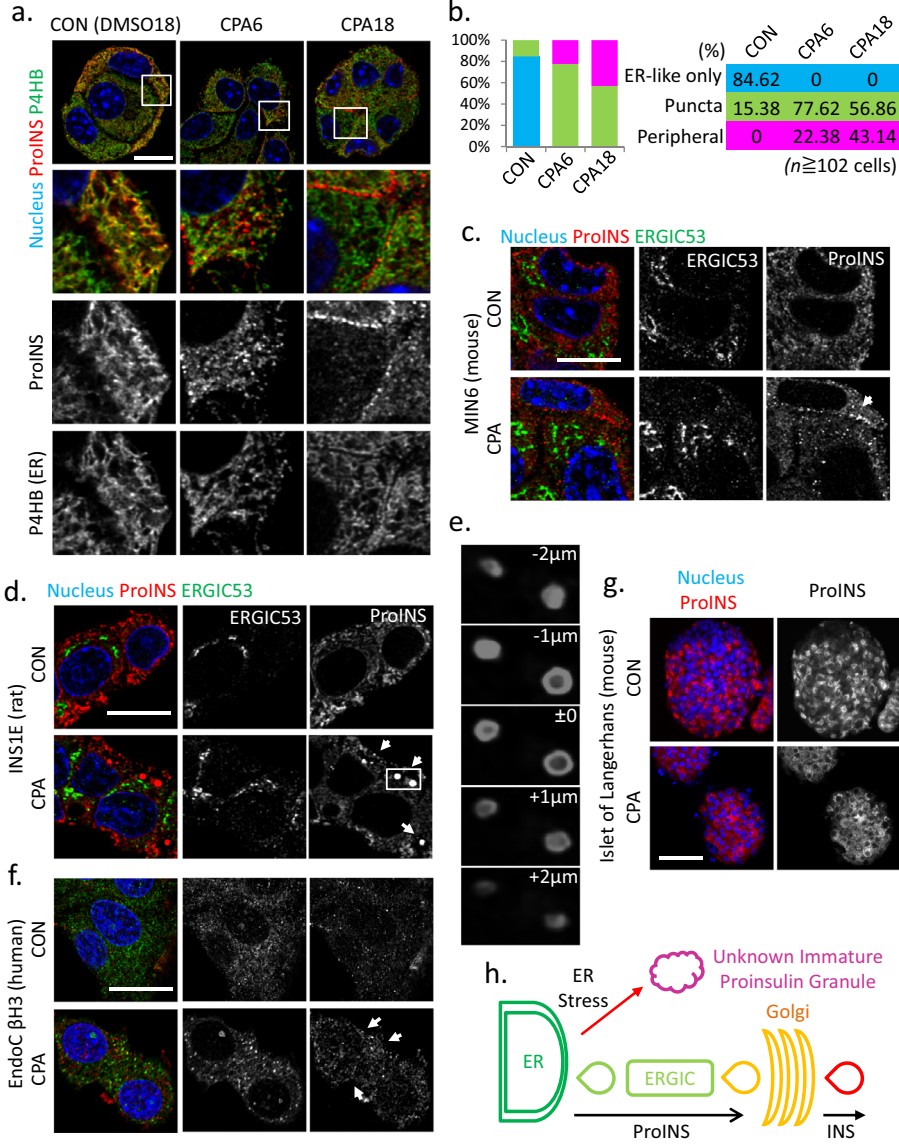

**Fig. 2 | Accumulation of cytoplasmic proinsulin granules in β-cells during chronic ER stress is conserved in rat and human species.** Fluorescence immunocytochemistry of MIN6 (**a**), INS1E (**d**), mouse islets (**g**) and EndoC βH3 (**f**), cells treated with CPA for 6 h and 18 h (MIN6), 6 h (INS1E) 18 h (mouse islets) and 72 h (EndoC βH3) for the indicated (color-coded) proteins. Boxes indicated the enlarged regions of images below each panel. **b** Quantitative analysis of the proinsulin puncta subcellular localization in the indicated treatments of MIN6 cells. **c** Fluorescence immunocytochemistry of MIN6 cells for the indicated proteins and treatments. **e** Reconstruction of stacking images represented by the box in **d**. **h** Schematic representation of the non-canonical exit of proinsulin from the ER. *n*, indicates number if cells from 3 independent experiments. The % of cells for the indicated phenotype are given in 2b in colored boxes and columns. Scale bar, 10 μm (**a**), 20 μm (**c**, **d**, **f**), 50 μm (**g**).

downregulation of genes involved in Maturity Onset Diabetes of the Young (MODY), proinsulin expression, synthesis, processing, and secretion. MEFs did not show differential expression of these genes expressed during chronic ER stress[22]. In addition, β-cells induced expression of more than half of genes known to participate in ER protein processing (133) under chronic adaptation conditions. We define this subset of genes as the β-cell-specific ER adaptome. Upon relief from stress, β-cells were able to regain expression of mature identity markers, revealing the resilience of β-cells to near lethal ER stress. Remarkably, upon iterative cycles of ER stress, β-cells lost this plasticity. Laser capture-microdissected islets[30,31] and single-cell RNA sequencing (scRNA-seq) data from T1D patients highlighted gene expression changes in β-cells of T1D patients, in agreement with chronic ER stress adaptive exhaustion and deregulation of the β-cell ER adaptome. We propose cycles of ER stress exhaust the β-cell's adaptive plasticity and promote an irreversible loss of function that contributes to T1D pathogenesis. We call this novel mechanism the 'β-cell exhaustive adaptive response (βEAR).

# Results

## Adaptation of MIN6 cells to chronic ER stress sustains insulin levels

MIN6 cells display β-cell-like insulin biosynthesis and secretion in response to glucose[32,33]. To test the effect of severe ER stress on MIN6 cell survival and on proinsulin synthesis, MIN6 cells were treated with CPA[34], which disrupts $Ca^{2+}$ homeostasis in the ER to induce the UPR[35]. As in MEFs[22], CPA treatment in MIN6 cells induced a time-dependent chronic ER stress response. The expression of hallmark proteins of the UPR, ATF4 and GADD34, showed a gradual increase (Fig. 1a), consistent with protein synthesis recovery[36]. Similar to MEFs[22], MIN6 cells showed inhibition of protein synthesis followed by partial recovery (Fig. 1b) without recovery of eIF2B activity (Fig. 1c).

The resumption of growth following recovery from 18 h of stress confirmed an adaptive stress response at that time point (Fig. 1d). The ability of MIN6 cells to recover from chronic stress was also accompanied by protein synthesis recovery (Fig. 1e) and restoration of eIF2B activity (Fig. 1c). Similarly, relief from ER stress inactivated stress-response signaling and stress-induced gene expression (Fig. 1f), suggesting a return to homeostasis[22,37]. We conclude MIN6 cells and MEFs have the same core mechanisms of regulating global protein synthesis during adaptation to chronic ER stress[22].

In MEFs, we have shown adaptive translational reprogramming involves a switch from eIF4E-dependent to eIF4E-independent, eIF3-dependent mRNA translation[22]. Because the most efficient mechanism of mRNA translation initiation utilizes eIF4F, we proposed this switch allows inefficient translation of all mRNAs, specifically ER-associated mRNAs. However, we also found a subset of mRNAs encoding stress-induced and ER-synthesized proteins are translated during adaptive chronic ER stress[22]. Consistent with MEFs, we found accumulation of stress-induced proteins in MIN6 cells exposed to CPA was similar in control and eIF4E-depleted cells (Supplementary Fig. 1a), supporting eIF4E-independent reprogramming of β-cells during chronic ER stress.

We next determined the regulation of proinsulin gene (*Ins1* and *Ins2*) expression during the transition from acute to chronic ER stress. Levels of *Ins1* and *Ins2* mRNAs decreased in MIN6 cells during chronic stress, suggesting an adaptive response. This decrease was gradually reversed during recovery from stress (Fig. 1g), supporting the regulation of proinsulin gene expression as part of the adaptive response. We also examined the levels of mature insulin and proinsulin, but neither the insulin monomer nor native insulin hexamer levels were altered during ER stress and recovery (Supplementary Fig. 1b). In contrast to the decreased levels of *Ins2* mRNA, during adaptive chronic ER stress, proinsulin protein levels decreased dramatically early in the stress response, presumably due to eIF2α-p-dependent inhibition of proinsulin mRNA translation[38]. Subsequently, proinsulin protein levels returned to normal at ≥12 h, despite decreased transcript levels during the chronic adaptive stress period (Fig. 1h). To demonstrate the regulation of proinsulin synthesis during progression of β-cells from acute to chronic ER stress, we showed that proinsulin synthesis was repressed at 1 h and de-repressed at 18 h of CPA treatment (Supplementary Fig. 1c). Further, we demonstrated the efficiency of translation of proinsulin mRNA also decreased at 1 h and was recovered at 18 h of CPA treatment (Supplementary Fig. 1d). Finally, in order to determine if proinsulin processing in chronic ER stress is similar to that in control cells, we examined the subcellular localization of proinsulin during adaptive chronic ER stress and subsequent recovery. Interestingly, intracellular and cell periphery-localized proinsulin puncta were observed during stress, but not during recovery (Fig. 1i). The sequestration of proinsulin during adaptive chronic ER stress into intracellular structures suggests proinsulin processing may be inhibited[39,40]. Taken together, our results show MIN6 cells reprogram translation during chronic ER stress with reversible inhibition of protein synthesis and proinsulin mRNA levels.

## CPA alters intracellular localization of proinsulin

Due to the essential role of ER-Golgi transport in insulin maturation, we examined proinsulin localization in MIN6 cells following CPA treatment. In CPA-treated cells, proinsulin was associated with the ER, shown by the colocalization of proinsulin and P4HB, an ER-resident molecular chaperone. CPA treatment (6 h) induced punctate staining of proinsulin in most cells. While the majority of proinsulin still colocalized with P4HB, the fraction of proinsulin in puncta did not stain positive for P4HB (Fig. 2a, b), suggesting enrichment of proinsulin in granule-like structures outside the ER near the cell periphery. After 18 h treatment, more than a third of cells had significant proinsulin staining in these peripheral granule-like structures, suggesting proinsulin largely migrated from the ER and became stable in these structures, likely

due to decreased processing. We further tested this hypothesis by comparing the degradation of proinsulin in non-stressed MIN6 cells and cells exposed to chronic ER stress via treatment with the protein synthesis inhibitor cycloheximide (CHX). Proinsulin in control cells is very unstable;[41] in contrast, we found proinsulin was very stable during chronic ER stress (Supplementary Fig. 1e). Furthermore, proinsulin degradation was insensitive to proteasome inhibition. Finally, functional recovery of CPA-treated MIN6 cells following CPA washout indicated partial recovery of the CPA-inhibited glucose-stimulated insulin secretion (GSIS) (Supplementary Fig. 1f). Taken together, we posit the resilience of β-cells to chronic ER stress includes positive and negative mechanisms of regulation of proinsulin gene expression.

Because plasma proinsulin is a risk factor for diabetes[42], we also determined if proinsulin is secreted from MIN6 cells treated with CPA[43] (Supplementary Fig. 2a, b). We show proinsulin is secreted during treatment with CPA (1 h and 18 h) and secretion decreases following washout of CPA from cells (after 18 h treatment). These data support ER stress-mediated accumulation of proinsulin in β-cells can lead to its secretion.

We next examined whether proinsulin colocalized with the ER-Golgi intermediate compartment (ERGIC), a transient intracellular structure between ER and Golgi[44]. We found no colocalization between proinsulin punctate staining and the ERGIC marker ERGIC53 in MIN6 cells in CPA-treated conditions (Fig. 2c). Next, we examined proinsulin and ERGIC53 in INS1E, a rat insulinoma cell line, and EndoC βH3, a human β-cell line. Large vesicle-like proinsulin-enriched structures appeared following 6 h of CPA treatment in INS1E cells (Fig. 2d, e). This early response of INS1E cells is consistent with the higher sensitivity of INS1E cells to ER stress. In human EndoC βH3 cells, CPA-induced punctate proinsulin immunostaining was observed near the cell periphery after 72 h of treatment and also did not colocalize with ERGIC53 (Fig. 2f). Similar proinsulin puncta at the cell periphery were observed in β-cells in mouse islets treated with CPA for 18 h (Fig. 2g). Collectively, our data suggest during ER stress adaptation, proinsulin maturation is impaired and proinsulin accumulates in structures outside the ER (Fig. 2h).

## MIN6 cells attenuate expression of genes associated with β-cell identity during chronic stress

The high demand for ER proteostasis in β-cells motivated us to examine β-cell-specific changes in gene expression in MIN6 cells during the progression from acute to chronic stress (Supplementary Fig. 3a, b). To this end, we analyzed effects of ER stress on genome-wide perturbations in transcriptomes and translatomes using total RNA sequencing (RNA-seq) and ribosome profiling. We analyzed the data for the *Ins2* gene for validation (about 50% of the mRNA in β-cells maps to *Ins* genes); while mRNA-seq reading frames in *Ins2* mRNA were unchanged, the distribution of ribosome footprints was enriched at reading frame 0 of the mRNA (Supplementary Fig. 3c), in agreement with ribosome footprints being at the ribosome P site[45,46]. Data quality was further confirmed by the selective enrichment of Ribo-seq reads on *Ins2* mRNA within the coding region, in contrast to the mRNA-seq reads distributed throughout the gene (Supplementary Fig 3d). We showed a high correlation among independent experiments for both transcriptome and translatome datasets (Supplementary Fig. 3e). Finally, we confirmed UPR pathway genes were significantly upregulated in all independent experiments (Supplementary Fig. 3f), validated by qRT-PCR (Supplementary Fig. 3g) and Western blotting (Fig. 1a).

Next, we determined the regulation of the global transcriptome and translatome during acute and chronic ER stress. For the translatomes, we normalized ribosome footprints to mRNA-seq datasets and assigned the normalized values the term 'ribosome occupancy' (ribo$^{ocp}$), as an indicator of the translation efficiency. We detected 10,163 transcripts encoding 6,938 genes in both transcriptome and

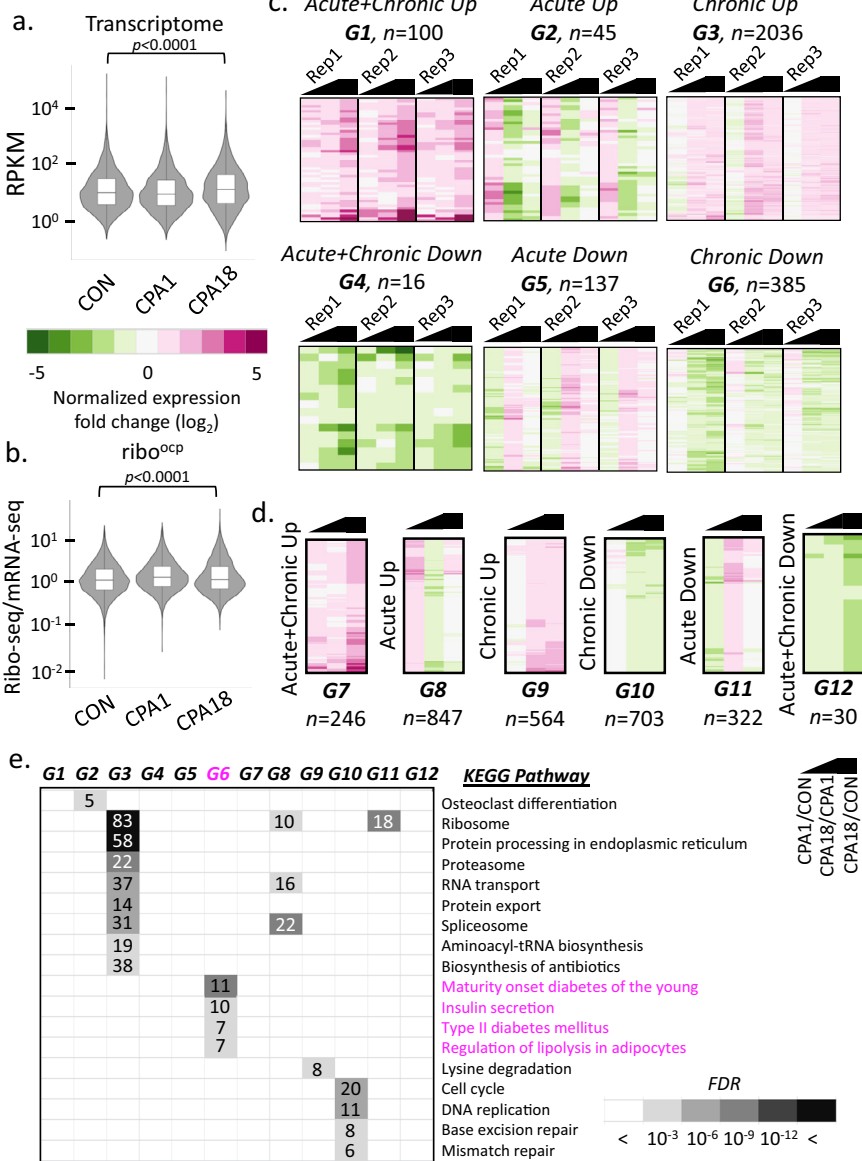

**Fig. 3 | Analysis of transcriptomes and translatomes during progression from acute to chronic phase of ER stress in MIN6 cells.** Violin plots representing transcriptomes (**a**) and translatomes (**b**), of untreated (CON), 1 h CPA treated (CPA1) and 18 h CPA treated (CPA18) cells. The boxes indicate the 25–75th percentiles, the midline indicates the median, whiskers show the maximum/minimum. *p*-value represents the statistical test by two-sided Wilcoxon signed-rank test. Heatmaps and the classifications of changes in the expression of transcriptomes (**c**) and translatomes (**d**) between CON and CPA1 (CPA1/CON), CPA18 and CPA1 (CPA18/ CPA1) and CPA18 and CON (CPA18/CON) datasets. Black symbol indicates the compared datasets. Three independent experiments were used with significance of gene expression. Letter code G1-G12 indicates the gene sets of assigned common regulation and *n*, indicates the number of genes per group. **e** KEGG pathway analysis of the classified groups of transcriptomes and translatomes. Boxed numbers indicate the number of genes identified in the individual pathways. *FDR*, False Discovery Rate.

translatome datasets. A significant change in mRNA abundance was observed in chronic (CPA18), but not acutely (CPA1) stressed cells compared to control cells (Fig. 3a), suggesting that reprogramming of the transcriptome is associated with adaptive chronic stress. We next assessed differences in riboᵒᶜᵖ in MIN6 cells treated with CPA (Fig. 3b). At CPA18, total riboᵒᶜᵖ decreased compared to the untreated control, suggesting reprogramming of mRNA translation. We then classified the differentially regulated transcripts into six groups by trends of changes in mRNA abundance (Fig. 3c) and riboᵒᶜᵖ (Fig. 3d) in response to CPA (Supplementary Data 1). Although a few mRNAs showed a transient increase or decrease in abundance in the acute response (G2 and G5), the predominant effect was observed in chronic stress, with 2036 genes being upregulated and 385 genes being downregulated compared to control cells (G3 and G6). Translational control in the

acute response was evident by 847 mRNAs being riboᵒᶜᵖ-up (G8) as compared to 246 mRNAs with continuous upregulation during progression of stress from acute to chronic (G7). In agreement with partial recovery of protein synthesis, 703 mRNAs decreased their riboᵒᶜᵖ during progression from acute to chronic ER stress (G10). To better understand establishment of adaptive proteostasis in chronic ER stress, we performed pathway analysis of these groups using the Kyoto Encyclopedia of Genes and Genomes (KEGG) database. Transcripts associated with ribosome biosynthesis, protein processing, degradation and transport, as well as RNA transport and splicing were upregulated during chronic ER stress. In contrast, mRNAs related to the ribosome, cell cycle, and DNA replication were markedly decreased in the acute or chronic stress states (Fig. 3e). Importantly, transcripts associated with β-cell identity (*Nkx2-2, Pdx1, Glut2, Ins1,* and *Ins2*) and

function (*Glut2*) were substantially decreased between CPA1 and CPA18. These results suggest MIN6 cells can withstand acute ER stress and maintain β-cell-specific gene expression. However, β-cell transcripts are progressively reduced upon establishment of adaptive homeostasis. These data further suggest loss of β-cells in diabetic islets may be caused by adaptive responses to chronic ER stress without recovery. We refer to this cellular response as the "β-cell exhaustive adaptive response" to chronic ER stress, or βEAR.

### Transcriptome reprogramming during chronic stress involves impaired expression of β-cell genes

To further assess how temporal changes of reprogramming relate to β-cell identity, we performed additional analysis of transcriptomes and translatomes of CPA1 (Fig. 4a) and CPA18 (Fig. 4b) vs control. In the acute response, we identified 798 mRNAs (11.5%) with increased and 300 mRNAs (4.32%) with decreased ribo$^{ocp}$, in agreement with translational reprogramming during acute stress. Changes in mRNA abundance were less prominent, with levels of 138 mRNAs increasing and 108 mRNAs decreasing (Fig. 4a). In contrast, there were more changes in mRNA abundance (1795) than translational control (1183) during the chronic response (Fig. 4b). It was notable, that changes in mRNA abundance increased from 1.99% in the acute phase to 20.41% in the chronic, while changes in ribo$^{ocp}$ were similar between the two. An early increase in mRNA abundance and translation was observed for *Atf4*, which was sustained during the chronic phase, but generally the percent/fold increase in ribo$^{ocp}$ was larger than the percent/fold increase in mRNA levels (Figs. 4c and 1a). Similarly, downstream of *Atf4*, *Chop* and *Gadd34* showed more than 2-fold increases in mRNA abundance and 3-fold increases in ribo$^{ocp}$ in the acute phase. We also examined changes in expression of 49 UPR-related genes (Fig. 4c and Supplementary Data 2). A shift in mRNA abundance was observed for most well-known UPR genes, as well as prosurvival genes, such as Wolframin ER Transmembrane Glycoprotein (*Wfs1*)[47]. In conclusion, β-cells show an enhanced response during the acute phase, with elevated mRNA abundance and ribo$^{ocp}$ for adaptation proteins.

The β-cell-specific transcriptome is important to maintain β-cell functions[48], misregulation of which can result in diabetes[49,50]. We found the mRNA abundance of β-cell-related genes, such as MAF BZIP Transcription Factor A (*MafA*)[51,52] and *Glut2* (Fig. 4d), a glucose transporter essential for glucose import in rodent β-cells[53–55], severely decreased during chronic ER stress and rapidly increased to pre-treatment levels after CPA washout (Fig. 4e, f). Since MAFA is a transcriptional activator of insulin[56], we expected the positive association observed between *MafA* and *Ins2* mRNA levels (Fig. 4g).

Genes associated with ER protein processing were upregulated in chronic stress, including those encoding molecular chaperones and factors involved in ER-associated degradation (ERAD) and ubiquitination (Fig. 4h). We identified an increase of VCP-interacting membrane selenoprotein (*Vimp*), Suppressor/Enhancer of Lin-12-like (*Sel1l*), and Derlin-2 (*Derl2*) in chronic ER stress, suggesting an active role for ERAD in the degradation of misfolded proinsulin[57]. We confirmed mRNA levels were increased by CPA treatment, and after washout, showed that *Sel1l* mRNA levels returned to basal levels (Fig. 4i). We further noticed that genes encoding ER molecular chaperones were differentially regulated during chronic ER stress and recovery from stress (Fig. 4j). Furthermore, we found chronic stress impaired both mRNA abundance and maturation of the convertase subtilisin/Kexin Type 2 (*Pcsk2*), a peptidase essential for insulin maturation (Fig. 4k, l). Because molecular chaperones are important in proinsulin disulfide bond formation and maturation, we investigated proinsulin misfolding in β-cells during ER stress. Using non-reducing SDS-PAGE gel electrophoresis, we found a gradual increase of high-molecular weight disulfide-linked proinsulin complexes upon CPA treatment; CPA washout was sufficient to

rescue this phenotype over time (Fig. 4m). Collectively, we show the balanced regulation of ER quality control genes results in reversible ER stress, but also compromises proinsulin gene expression and protein maturation.

### ER-Golgi transport in adaptation to ER stress in MIN6 cells

It is currently unknown whether secretory cells utilize specific cellular machinery to achieve adaptive homeostasis compared to non-secretory cells. Thus, we compared the transcriptomes of acute and chronic ER stress adaptation in MIN6 cells and MEFs. This analysis in MIN6 cells identified 1534 mRNAs with increased and 307 mRNAs with decreased abundance between the acute and the adaptive timepoints, respectively (Supplementary Fig. 4a). Similarly, we identified 603 and 296 mRNAs subjected to positive or negative translational control, respectively. mRNAs encoding proteins of the ribosome biosynthesis pathway were enriched in the positive group, while the ribo$^{ocp}$ of cell cycle related mRNAs was reduced. Analysis of pre-existing data from MEFs[22] identified 567 and 1247 mRNAs which showed upregulation or downregulation of mRNA abundance, respectively. KEGG pathway analysis of the downregulated genes between MIN6 cells and MEFs showed the stress responses were cell type-specific and tailored to cellular function: a decrease of β-cell-related functions was observed in MIN6 cells, while mRNAs of cell adhesion and ECM proteins were decreased in MEFs during the transition from acute to chronic stress adaptation (Supplementary Fig. 4b).

In corroboration with common adaptation mechanisms[58], we identified an enrichment of genes in the ER protein processing pathway among mRNAs with increased abundance (Supplementary Fig. 4c). 68 mRNAs showed increased abundance and 10 mRNAs showed translational upregulation of this cellular pathway in MIN6 cells. In MEFs, a smaller proportion of mRNAs (37 of 78), were regulated, suggestive of similar, but less elaborate adaptation machinery. We analyzed the 37 overlapping genes between MIN6 and MEFs by Gene Ontology (GO)[59], and observed these genes were highly associated with ER stress amelioration, including molecular chaperones and ERAD. The 41 MIN6-specific genes were enriched for functions in ER-to-Golgi vesicle transport (Supplementary Fig. 4d). This analysis revealed a unique adaptive event in MIN6 cells, i.e. an enhancement of the first step of vesicular transport, suggesting stress adaptation is influenced by cellular function. We termed this group of 78 genes in MIN6 cells the 'β-cell-specific adaptome'.

### Lysosome function is impaired during chronic ER stress

Formation of proinsulin granules was reversible, supporting their formation as an adaptive response to chronic ER stress (Fig. 1i). To shed light on the mechanism of proinsulin accumulation, we examined gene regulation in chronic ER stress (Fig. 5a), and noted that genes encoding subunits of the proteasome were upregulated (Fig. 5b). By pathway analysis of the 497 downregulated mRNAs, we found lysosome biogenesis genes, such as *Arsb*, a lysosomal arylsulfatase, and *Lamp2*, a lysosomal associated membrane protein, were significantly repressed (Fig. 5c), suggesting impaired lysosome function. Accordingly, we observed a time-dependent reduction of lysosomal enzymatic activity during CPA treatment which gradually returned to levels of untreated cells 6 h after CPA washout (Fig. 5d). We also monitored the protein levels of the autophagy substrate ubiquitin-binding protein p62 (*Sqstm1*), and the conversion of microtubule-associated protein 1 A/1B-light chain 3 (LC3-I) to LC3-II in MIN6 cells. We observed a gradual increase of p62 and LC3-II levels during CPA treatment (Fig. 5e). Lastly, we observed an increase of punctate LC3 staining in MIN6 cells after 18 h CPA treatment (Fig. 5f). Taken together, we conclude the pathway of autophagosome-to-lysosomal degradation was inhibited during chronic ER stress in MIN6 cells.

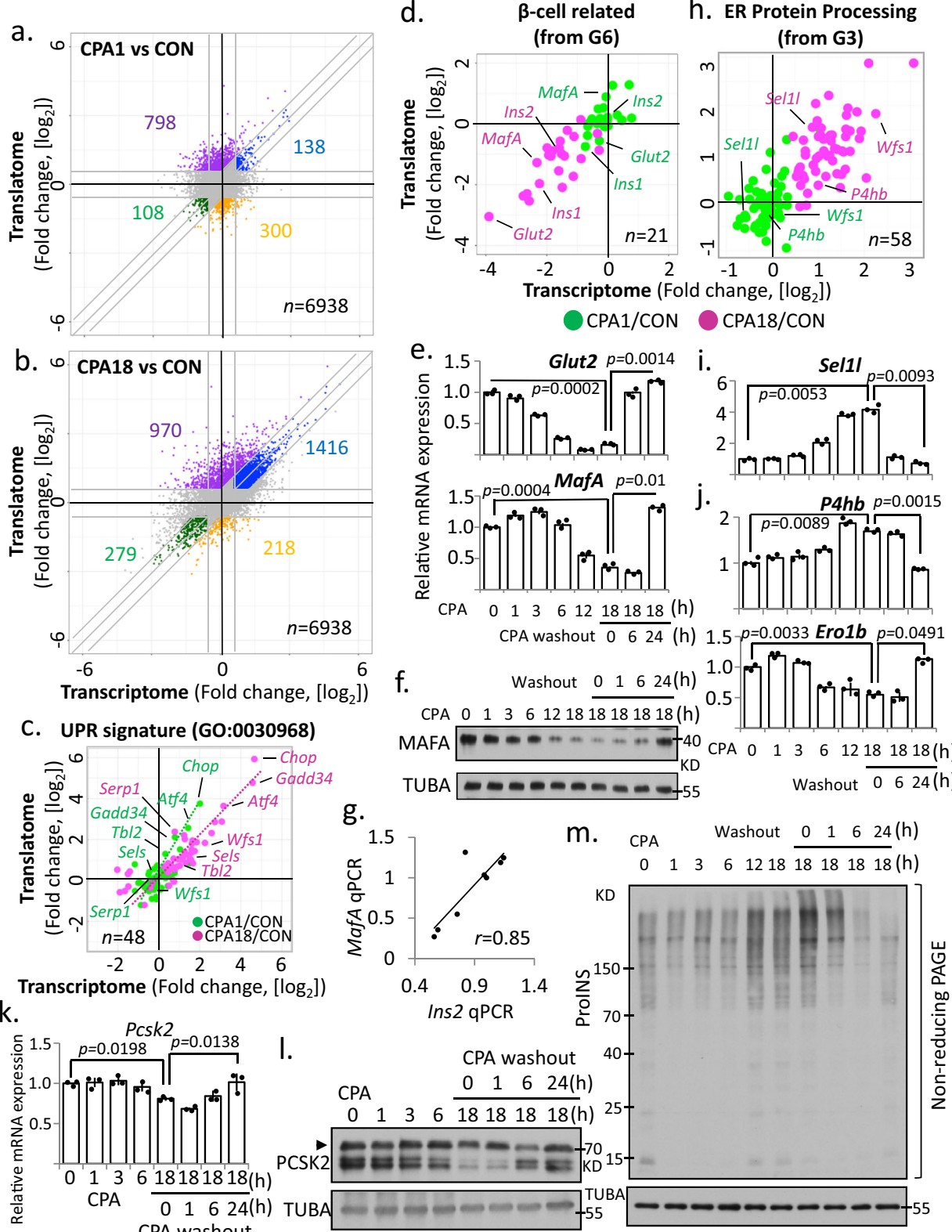

### Impairment of lysosome activity correlates with misfolded proinsulin accumulation

Our analysis demonstrated lysosomal activity gradually declined during chronic ER stress in MIN6 cells, while the ubiquitin-proteasome system (UPS) was upregulated (Figs. 3e, 4h and 5c). Moreover, we revealed an accumulation of misfolded proinsulin outside the ER upon CPA treatment (Figs. 2 and 4m), implying

disrupted proinsulin proteostasis in chronic stress. To determine if the UPS or the lysosome were involved in this regulation, we treated MIN6 cells with CPA for 18 h together with proteasome inhibitor MG132 or Baf A1 and monitored proinsulin folding status. Lysosomal impairment by Baf A1 over the last 4 h of the 18 h of CPA treatment strongly increased the amount of misfolded proinsulin (Fig. 5g). Moreover, CPA washout in the presence of Baf A1 led to

**Fig. 4 | Transcriptional and translational reprogramming in MIN6 cells in response to CPA induced ER stress.** Scatterplots of fold changes in CPA1 vs CON (**a**) and CPA18 vs CON (**b**). **c, h** Scatterplot of fold changes of the UPR signature (GO:0030968) gene expression between acute (CPA1/CON) and chronic (CPA18/CON) ER stress. **d** Scatterplot of fold changes of genes in groups identified in Fig. 3E (G6 and G3) in acute (CPA1/CON) and chronic (CPA18/CON) ER stress. **e, i, j** qRT-PCR analysis for the indicated mRNAs normalized to *GAPDH* mRNA levels, in MIN6 cells treated with CPA for the indicated times, *n* = 3. **f, l** Western blot analysis for the indicated proteins in MIN6 cells. **g** Scatterplot representing an association between *Ins2* and *MafA* mRNA levels measured by qRT-PCR. **k** qRT-PCR analysis of *Pcsk2* mRNA levels normalized to *GAPDH* mRNA levels, in MIN6 cells under the indicated treatments, *n* = 3. **m** Western blot analysis of proinsulin in non-reducing SDS-PAGE electrophoresis. Error bars represent S.E.M. *p*-value represents the statistical test by two-tailed paired Student's *t*-test. Representative western blotting images were shown. Dots in all plots represent independent experiments. Source data are provided as a Source Data file.

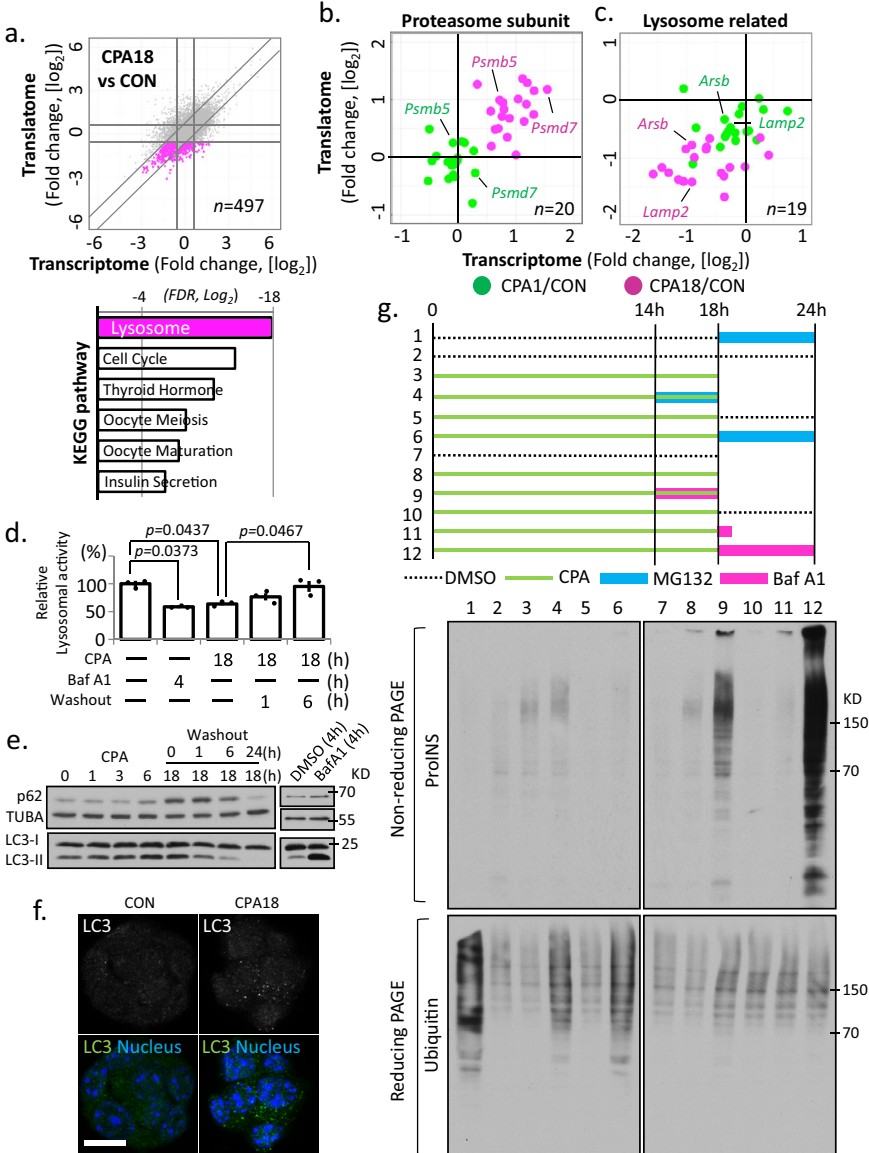

**Fig. 5 | Impairment of lysosome activity correlates with misfolded proinsulin accumulation during chronic ER stress. a** Scatterplot of fold changes in CPA18 vs CON MIN6 cells, with the repressed gene expression indicated in pink, and the KEGG pathway analysis of the highlighted group, below **a**. Scatterplots of fold change in acute (CPA1/CON) and chronic (CPA18/CON) ER stress for genes related to the proteasome (**b**) and the lysosome (**c**). **d** lysosome enzymatic activity measured in MIN6 cells treated with Baf A1 or CPA. *n* = 3 independent experiments. Error bars represent S.E.M. *p*-value represents the statistical test by two-tailed paired Student's *t*-test. Source data are provided as a Source Data file. **e** Western blot analysis of the indicated proteins in MIN6 cells. **f** Fluorescence immunocytochemistry for LC3, in MIN6 cells treated with CPA for 18 h. Nuclei are indicated with staining in blue. Scale bar is 20 μm. **g** *upper*, schematic representation of 12 different MIN6 treatments with CPA, MG132 or Baf A1. Green lines indicate CPA treatment. CPA was washed out between 18–24 h; *lower*, Western blot analysis for proinsulin by non-reducing PAGE electrophoresis in extracts prepared from the 12 MIN6 treatments. Western blot analysis for ubiquitin by reducing PAGE electrophoresis.

dramatic accumulation of misfolded proinsulin, while MG132 had only a modest effect. Therefore, our data suggest CPA-mediated chronic ER stress impairs clearance of misfolded proinsulin by disrupting autophagosome-to-lysosomal degradation.

**Chronic ER stress adaptation genes are downregulated in T1D**

Pancreatic β-cells encounter numerous microenvironment changes over their life span, and studies have highlighted the importance of the ER stress response in diabetes[17,60]. We hypothesized that β-cell loss/

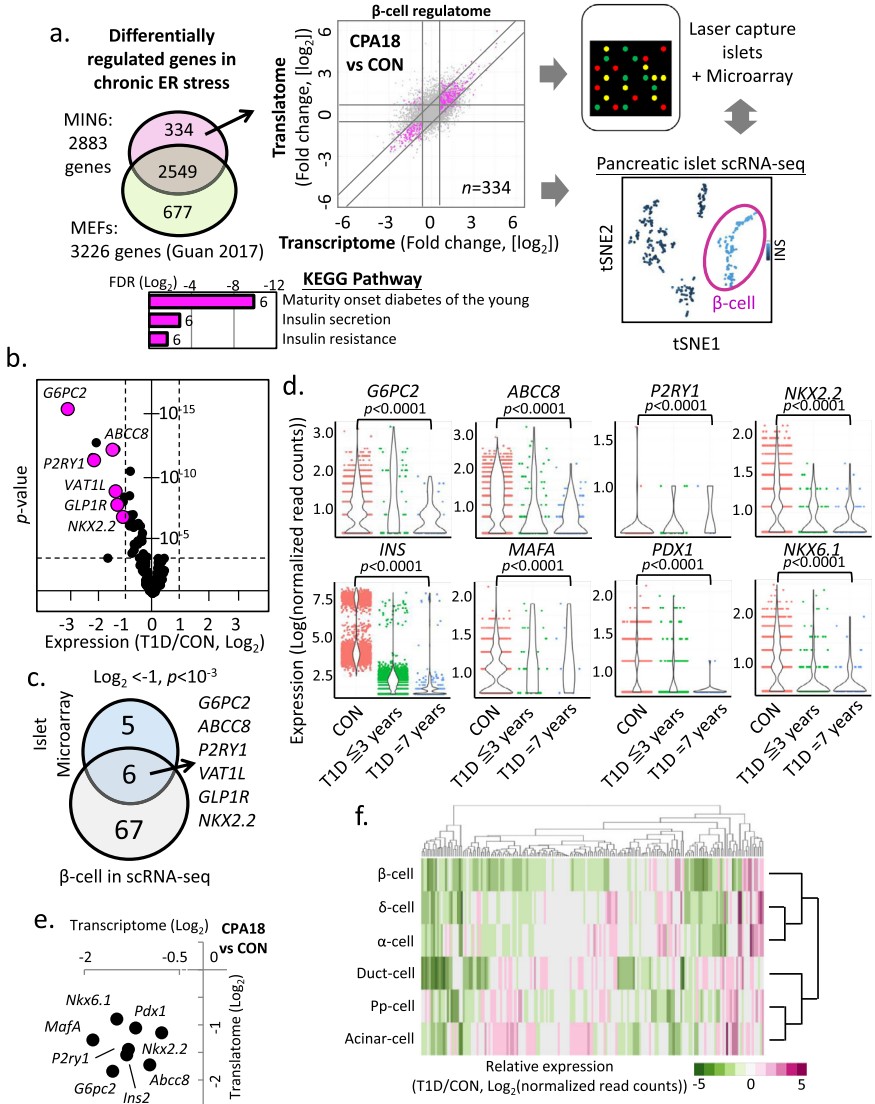

**Fig. 6 | Altered expression of the β-cell ER stress-induced gene set (regulome) in T1D islets reveals biomarkers of T1D progression. a** Schematic representation of the β-cell ER stress gene set (334 genes) in MIN6 cells and determination of its expression in T1D islets or scRNA-seq datasets. Scatterplot indicates in color the expression patterns of the 334 gene set during chronic ER stress in MIN6 cells. The KEEG pathway of the 334 gene set is shown below **a. b** The 334 gene set expression profile in T1D islet microarray datasets, shown by a volcano plot. *p*-value represents the statistical test between CON and T1D by two-sided Wilcoxon–Mann–Whitney test. **c** Venn-diagram showing the common repressed genes of the 334 gene set in T1D, between islet microarray and β-cells scRNA-seq datasets. *p*-value represents the statistical test between CON and T1D by two-sided Wilcoxon–Mann–Whitney test. **d** Expression of genes in β-cells scRNA-seq datasets of healthy donors (CON), early T1D (T1D ≦3 years) and prolonged T1D (T1D = 7 years) patients. *p*-value represents the statistical test by two-sided Wilcoxon–Mann–Whitney test. **e** Scatterplot of fold change in MIN6 cells during chronic ER stress (CPA18 vs CON). **f** Heatmap of gene expression profiles of the 334 gene set in different cell types of T1D scRNA-seq datasets, compared to healthy donors.

dysfunction in diabetes could be associated with the homeostatic state of ER stress adaptation or its failure. To test this hypothesis, we selected genes that were affected in MIN6 cells during the chronic adaptive state, and filtered out those that exhibited similar behavior in MEFs[22]. This allowed us to establish a β-cell-specific ER stress gene dataset. This gene set, which we called the 'regulome', contains 334 genes enriched for β-cell-associated functions by KEGG pathway analysis (Fig. 6a). We examined the expression levels of this gene set in a pre-existing microarray database (Supplementary Data 4), generated from laser-excised islets[30,31] and the scRNA-seq data produced by the Human Pancreas Analysis Program (HPAP)[61] from patients with T1D and T2D and healthy donors (Fig. 6b). We identified 11 genes that showed downregulation in T1D samples (Fig. 6b), but not in T2D samples (Supplementary Fig. 5a). One gene, *PPP1R1A*, was identified in both T1D and T2D, but showed less significant changes in T2D. 10 of

the 11 downregulated genes were present and decreased in MIN6 cells during chronic ER stress adaptation.

Pancreatic islets comprise several cell types, including α-cells (30–50%), β-cells (50–60%), δ-cells (<10%), pp-cells (<5%) and ε-cells (<1%). In T1D patients, there is a specific loss of β-cells;[62,63] thus, the downregulation of the 11 downregulated genes in the T1D microarray dataset could be a result of differential expression in multiple islet cell types. To further investigate β-cell-specific responses in T1D, we first compared the expression profiles of healthy donor and T1D patient islets[61] for the 11 genes in the scRNA-seq dataset (Supplementary Fig. 5b). We found that the expression of all 11 genes was decreased in T1D β-cells (Supplementary Fig. 5c), and using a more stringent cut off (FDR≤10⁻³), 6 of the 11 genes were significantly downregulated in β-cells of T1D patients (Fig. 6c, d). In addition to these 11 genes, *INS* and *MAFA* mRNA abundance was also decreased in β-cells of patients with

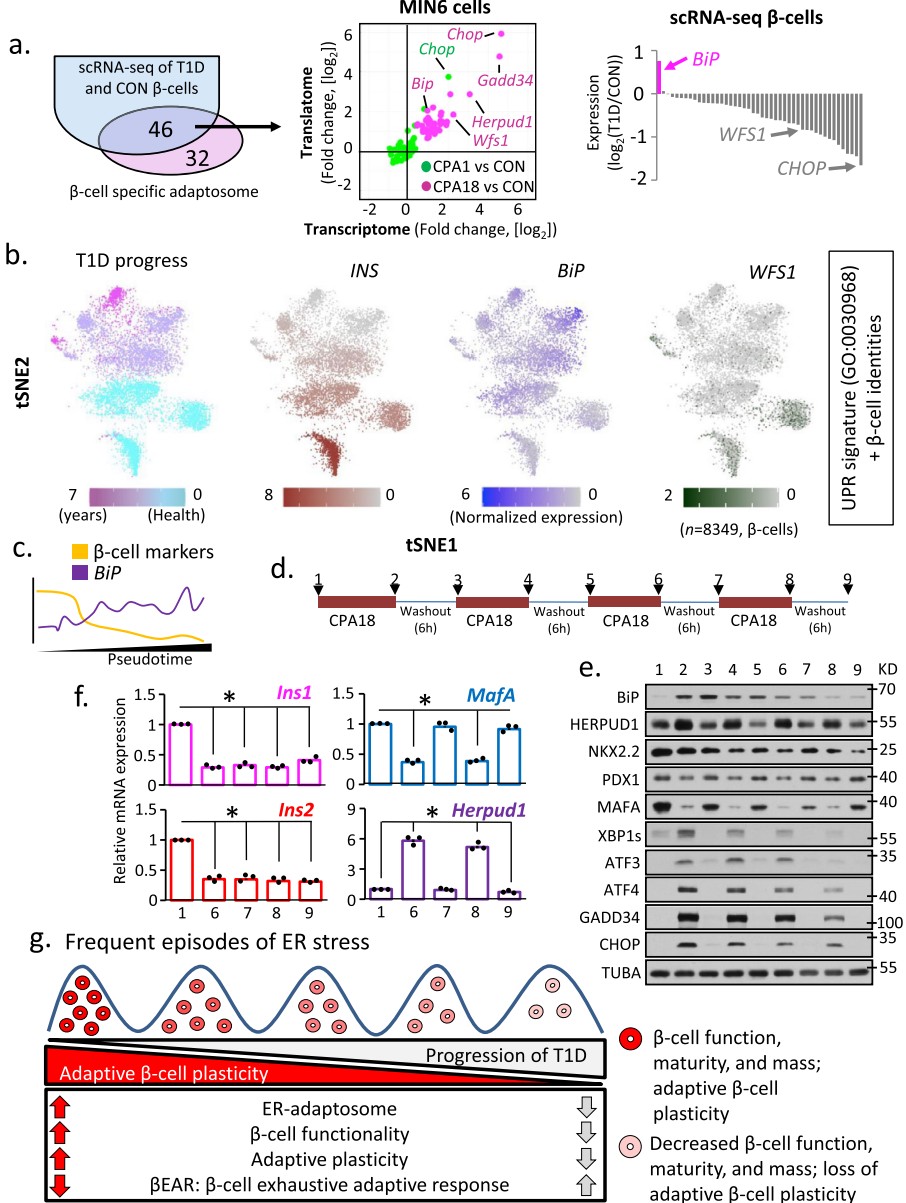

**Fig. 7 | Progression of T1D pathogenesis correlates with ER-stress-induced adaptive exhaustion and suppression of β-cell maturity markers. a** Schematic representation of the common genes between the β-cell specific adaptome in MIN6 cells (78 genes) and the β-cells scRNA-seq datasets. Plots show the regulation of the 46 genes in MIN6 cells during acute and chronic ER stress (middle) and β-cells scRNA-seq from healthy and T1D patients (right). **b** Relevant distance in gene expression between two data sets: (tSNE1) scRNA-seq dataset from all human β-cells (healthy and T1D, 8349) and (tSNE2) the expression of a group of genes consisting of both, the UPR signature genes (GO:0030968), and 6 β-cell-specific gene markers (INS, PAX6, NKX2.2, NKX6.1, MAFA and PDX1). Colored from left to right, T1D progression, *INS*, *BiP* and *WFS1*. Darker color indicates higher expression of the respective genes. Raw data were normalized as described in Materials and Methods. **c** pseudotime analysis of tSNE data from b, showing decrease in β-cell markers and increase in *BiP*. **d** Schematic of stress/recovery cycles and Western blot analysis of UPR and β-cell identity genes. **e** Western blot analysis of the indicated proteins in MIN6 cell extracts isolated from the time points indicated in **d**. **f** qRT-PCR analysis of RNA levels normalized to *GAPDH* mRNA, for *Ins1*, *Ins2*, *MafA* and *Herpud1* mRNAs in the last 2 stress/recovery cycles. *n* = 3 independent experiments. Error bars represent S.E.M. All comparisons were to sample 1. *p*-value represents the statistical test by two-tailed paired Student's *t*-test. Dots in all plots represent independent experiments. Individual *p*-values (*) and source data are provided as a Source Data file. **g** Working hypothesis model of βEAR in T1D pathogenesis.

T1D (Fig. 6d), while *G6PC2*, *ABCC8* and *NKX2.2* showed a significant decrease in expression compared to healthy donors (Fig. 6c). All 8 genes shown in Fig. 6d were also downregulated in MIN6 cells during chronic ER stress (Fig. 6e). Lastly, we examined how the regulome was regulated in the scRNA-seq dataset of T1D islets (Fig. 6f). We observed a larger decrease in expression of these genes in β-cells in comparison to other cell types in T1D (β-cells/all other cell types = 57.23%/29.13 ± 0.79%). Only 7.51% of the gene set showed an increase in β-cells, lower than other cell types (12.95 ± 2.23%). Collectively, our analysis revealed reduced abundance of distinct mRNAs of β-cell chronic ER

stress genes in T1D. This finding supports the hypothesis that chronic ER stress adaptation likely fails in β-cells in T1D, leading to targeted downregulation of β-cell function- and identity-specific mRNAs. We propose failure to adapt to chronic ER stress, or βEAR (Supplementary Data 3), may contribute to the pathogenesis of T1D.

## Cycles of chronic ER stress lead to decreased plasticity and loss of β-cell-specific gene expression

Decreased expression of β-cell-specific genes in T1D islets and β-cells from T1D patient islets suggested inadequate adaptation to chronic ER

stress. To test this, we compared the expression profile of the MIN6 β-cell-specific adaptome (78 genes) with scRNA-seq data from T1D and healthy donor β-cells. We found overlap of 46 of these genes expressed in both healthy donors and T1D patients. In contrast to upregulation during chronic adaptation in MIN6 cells (Supplementary Fig. 4c), most adaptome genes were downregulated in T1D patients compared to healthy donors (Fig. 7a). Notably, the expression of the UPR sensor BiP[64], was significantly increased in T1D patients. Strikingly, the relative expression of these genes in α- and δ-cells is different than in β-cells; many genes were upregulated in α- and δ-cells, including BiP, suggesting partial adaptation to ER stress in these cell types (Supplementary Fig. 6a, b). Our data indicate β-cells in T1D subjects are likely more sensitive to ER stress and lose their adaptive ER proteostasis faster than other cell types during disease progression.

Next, we looked at the relevant distance in gene expression between (i) the scRNA-seq dataset from all human β-cells (healthy and T1D, 8,349) and (ii) the expression of UPR signature genes (GO:0030968)[29] and β-cell-specific gene markers (INS, PAX6, NKX2.2, NKX6.1, MAFA, and PDX1). Similarity was determined via the t-distributed stochastic neighbor embedding algorithm, or t-SNE[65](Fig. 7b). This analysis identified clusters of β-cells with the UPR signature but heterogeneous UPR marker gene expression. By mapping the T1D disease progression (Supplementary Fig. 5b) on the UPR signature clusters of β-cells, we identified clusters corresponding to healthy donors and others derived from patients with more advanced T1D (Fig. 7b). This further supported differential clustering of gene expression patterns for healthy and T1D subjects (Supplementary Fig. 6c). We then determined the expression of INS mRNA in each cluster to define β-cell identity, and found good correlation between high levels of INS expression and healthy donors. We also noticed a decline of INS mRNA abundance with T1D pathogenesis and an anti-correlation between INS and BiP gene expression with T1D pathogenesis (Fig. 7b). In agreement with unresolved ER stress, expression of the β-cell adaptome marker WFS1 was decreased in T1D pathogenesis (Fig. 7a, b), and was more abundant in the healthy β-cell clusters. Finally, in order to define the maturity of β-cells in T1D subjects, we performed pseudo-timing analysis by in silico pseudo-time reconstruction in single-cell RNA-seq analysis using TSCAN[66], and Monocle 3[67] packages in R. We showed a gradual decrease of β-cell markers and increase of BiP (Fig. 7c). Taken together, these data suggest β-cells from T1D patients experience an unresolved UPR.

It has been recently reported that heterogeneity in β-cell maturity in islets is essential for their function in vivo[68]. Thus, our data showing ER stress decreases expression of genes involved in β-cell maturity, combined with the documented heterogeneity of β-cells in islets in vivo, suggests that β-cells undergo cycles of ER stress and recovery in response to metabolic stress. We therefore reasoned the collapsing chronic ER stress adaptation response in T1D islets could result from exposure to frequent ER stress cycles or after prolonged ER stress. We therefore tested these two hypotheses in MIN6 cells. The UPR markers showed complete reversibility after prolonged stress (Supplementary Fig. 7a, b). Although prolonged ER stress decreased cell survival (Supplementary Fig. 7c, d), the surviving cells did not indicate increased cleaved caspase 3 (Supplementary Fig. 7e). Our data suggest β-cells can endure prolonged ER stress and still recover.

We next determined the effect of multiple cycles of recovery from chronic ER stress exposure on the levels of stress-induced and β-cell specific proteins. MIN6 cells were challenged with four cycles of 18 h CPA treatment followed by 6 h recovery in fresh medium (Fig. 7d). We showed an increase of cell loss with cycle number (Supplementary Fig. 7f); however, no cleaved Caspase 3 was detected at the end of the fourth stress cycle, suggesting the attached cells were resilient to cyclic ER stress (Supplementary Fig. 7g). We next examined the expression of UPR genes and β-cell identity markers in response to the stress/recovery cycles. We found a gradual reduction of UPR proteins with

increasing cycle number. Expression of BiP was induced in the first cycle of CPA treatment, but expression was not increased in subsequent cycles (Fig. 7e). In agreement with basal UPR being essential for β-cell health[69], BiP and XBP1s were expressed in control MIN6 cells (Fig. 7e). Although these data suggest BiP regulation is lost with stress/recovery cycling, protein levels remained higher in cycling than in untreated cells (Fig. 7e). Notably, the β-cell marker NKX2.2 showed a decrease and partial recovery during the second and third stress/recovery cycle, but was unchanged following the first cycle, and NKX2.2 protein levels gradually decreased with cycles (Fig. 7e). MAFA, on the other hand, was affected by early stress cycles; MAFA showed a significant decrease, with partial recovery at the end of the first and subsequent cycles (Fig. 7e). Lastly, we found a significant irreversible reduction in Ins1 and Ins2 gene expression in the last two stress cycles (Fig. 7f). The pattern of changes in MafA and Herpud1 mRNA levels in the last two cycles were in good correlation with the changes in their protein levels (Fig. 7e, f). Taken together, these results show increasing frequency of stress/recovery cycling progressively diminish β-cell plasticity and the return to homeostasis. Since T1D is a condition of increased β-cell sensitivity to environmental and nutritional factors, we propose frequent episodes of chronic ER stress in β-cells may contribute to diminution of β-cell identity and promote β-cell loss (Fig. 7g).

Although we focused on molecular adaptive plasticity here, metabolic plasticity is also expected to be part of the adaptation mechanism to chronic ER stress[70,71]. We found biosynthesis of cholesterol, owing to its importance for GSIS[72], decreased during chronic ER stress and gradually recovered upon stress removal (Supplementary Fig. 8a); the inhibition of cholesterol biosynthesis was consistent with the adaptive response of β-cells to chronic ER stress. While Ins mRNA levels decrease, mechanisms of inhibition of secretion of proinsulin or insulin such as decreased cholesterol biosynthesis can generate a reserve pool of insulin for recovery from stress.

## Discussion

T1D is preceded by β-cell dysfunction in islets, but the mechanisms are not well understood[73,74]. ER stress could cause β-cell dysfunction and progression of T1D. In the T1D microenvironment, ER stress can develop via several factors inherent to β-cells or in response to cytokines secreted by local T cells and macrophages[74]. Here, we tested the intrinsic effect of ER stress and the impact of recovery from stress on β-cell identity as a means to understand the pathogenesis of T1D. Using a chemical that reversibly elicits an ER stress response, we revealed plasticity of β-cells during their response to chronic stress. We showed during chronic stress, β-cells undergo a homeostatic change associated with reprogramming of their transcriptome and translatome. Chronic stress altered more than half of genes known to participate in ER protein processing, while also compromising β-cell identity. Upon relief from stress, β-cells regained their mature identity, indicating plasticity. This suggests that episodes of ER stress, even at high amplitude, are tolerated by β-cells in a manner designed to protect ER function.

Importantly, frequent repeated cycles of ER stress diminished β-cell plasticity, as shown by an inability to recover expression of their identity genes, including Ins1 and Ins2. Laser-captured islets and scRNA-seq data from patients with T1D indicated gene expression alterations in T1D β-cells consistent with a model of ER stress adaptive exhaustion and deregulation of the β-cell ER adaptome during disease progression. Thus, it appears that frequent cycles of ER stress over time exhaust the adaptation machinery of β-cells to promote an ultimately irreversible loss of function that likely facilitates T1D pathogenesis.

Our findings on adaptive plasticity to chronic ER stress are reminiscent of recent studies on β-cell heterogeneity in vivo, as a requirement for islet function[68,75]. It was shown in a mouse model that adult islets are composed of mature and immature β-cells, and the balanced

expression of these β-cell populations was important for the physiological metabolic response of the islets[68]. The authors indicated that β-cell isolation from islets can eliminate differences in maturity. We can speculate that subpopulations of β-cells in islets represent cells in states of adaptation to ER stress (low levels of *MafA* and *Pdx1*) and recovery from ER stress (high levels of *MafA* and *Pdx1*). Therefore, the heterogeneity in β-cell maturity in normal islets may be a physiological response to cycles of ER stress and recovery in vivo.

In our experimental system of CPA-induced ER stress, proinsulin exited the ER and formed granule-like structures. The correlation with the decreased maturation of PCSK2, the protease that cleaves proinsulin to produce the mature insulin[76], and decreased lysosomal activity, a mechanism of clearance of insulin granules[77], may implicate these events as part of that mechanism. Additional possibilities could include changes in luminal acidification[78] and decreased cholesterol in immature secretory granule membranes[79]. Furthermore, dysfunction of the zinc transporter SLC30A8, a known autoantigen in T1D, can also impact proinsulin processing[80]. Interestingly, cholesterol biosynthesis (Supplementary Fig. 8a) and the RNA levels of SLC30A8 decreased during chronic ER stress in MIN6 cells, supporting multiple potential contributions to the inhibition of proinsulin processing.

Our studies in MIN6 cells can be considered a novel methodology to identify biomarkers for T1D and T2D pathogenesis. Most of the 11 genes identified here (Fig. 6) associate with T1D and T2D, suggesting a role of ER stress adaptation in the diabetic homeostatic state. Remarkably, seven of the 11 signature genes are associated with either insulin maturation or GSIS. PCSK1/3 and PCSK2 are involved in the C-peptide removal from proinsulin, a crucial step in insulin maturation. Deficiency of three plasma membrane proteins (GLP1R, ABCC8, and P2RY1) is tightly linked to abnormal GSIS and associated with diabetes[81–83]. Two secreted proteins, PPP1R1A and SCGN, also showed decreased levels in both ER stress adaptation in MIN6 cells and T1D patients. PPP1R1A is an inhibitor of the protein-phosphatase 1, and is positively correlated with insulin secretion and negatively with HbA1c. SCGN is an insulin-interacting calcium sensor protein that stabilizes insulin. Due to the secretion property of these two proteins, they have been proposed as biomarkers for diabetes detection[84,85]. SYT4, Synaptotagmin 4, localizes in mature insulin granules and is important for GSIS; SYT4 overexpression increases GSIS threshold, while ablation compromises GSIS[86]. Finally, we implicate a new diabetic candidate gene VAL1L, Vesicle Amine Transport 1-Like, associated with ER stress adaptation in MIN6 cells and T1D. Overall, our data suggest the genome-wide cellular response in MIN6 cells can lead to new discoveries in the pathogenesis in diabetes.

Although our current study modeled changes typical of pre-T1D, alternative experimental systems can study T2D pathogenesis. For example, lipotoxicity can induce ER stress in tissue cultured β-cells, and is a factor associated with β-cell loss in T2D. Similar to the studies presented here via the use of a chemical ER stressor, palmitic acid-induced ER stress in MIN6 cells results in a similar reversible decrease in MAFA protein levels (Supplementary Fig. 8b, c). Future studies can determine the commonalities between different physiological and pathological stressors and the bona fide ER stress cellular response described here.

One limitation of our study is that while MIN6 cells provide an excellent system to discover factors involved in the resilience of β-cells to chronic ER stress, mimicking the range of stressors in human diabetes that include hyperglycemia, hyperlipidemia, cytokine action, etc. or paracrine interactions in the islet environment is not possible. We showed that in mouse islets treated with CPA, levels of MAFA in β-cells decreased, followed by quick recovery when CPA was washed out (Supplementary Fig. 9a, b). Furthermore, GSIS was blunted in CPA-treated cells and resumed recovery when CPA was washed out (Supplementary Fig. 9c). Hence, future experiments mimicking the islet microenvironment in cultured mouse and human islets would provide invaluable insight into states of reversibility, regulation, and function of βEAR and pave the way for novel approaches preventing the loss of functional β-cell mass in T1D.

## Methods

### Cell lines, cell culture conditions
Mouse insulinoma MIN6 cell was purchased from AddexBio Technologies (Catalog no.C0018008). Human EndoC-βH3 was purchased from Human Cell Design. MIN6 cells were cultured in DMEM containing 4.5 g/L glucose, 10% heat-inactivated FBS, 1 mM sodium pyruvate, 0.1% β-mercaptoethanol, 2mM L-glutamine and 100 U/mL of Penicillin and Streptomycin. Rat insulinoma INS1E cells were cultured in RPMI 1640 containing 2 g/L glucose, 10% FBS, 10 mM HEPES, 0.1% β-mercaptoethanol, 2mM L-glutamine and 100 U/mL of Penicillin and Streptomycin. Human EndoC-βH3 cells were cultured in DMEM containing 5.6 mM glucose, 2% BSA fraction V, 10 mM nicotinamide, 5.5 µg/mL transferrin, 6.7 ng/mL sodium selenite, 50 µM 2-mercaptoethanol, 100 U/mL Penicillin, 100 U/mL Streptomycin and 10 µg/mL puromycin. Cells were cultured at 37 °C, supplied with 5% $CO_2$. MIN6 and INS1E cells were subcultured in a 1-to-4 ratio weekly. EndoC-βH3 cells were subcultured in a 1-to-2 ratio biweekly. For shRNA knockdown experiments, lentiviral particles expressing shRNA against the eIF4E mRNA was prepared and propagated in HEK293T cells as described previously[5] using the second generation pLKO.1, psPAX2 and pMD2.G vectors. After two rounds of lentiviral infection, cells were selected under puromycin (10 µg/mL) for three days.

### Chemicals, reagents and antibodies
Chemicals used in this study: CPA (200 µM, MIN6; 50 µM, INS1E; 500 µM, EndoC βH3) (Tocris). Met/Cys-deficient DMEM medium (Gibco). N-ethylmaleimide (NEM) (Sigma-Aldrich). Trans ³⁵S-amino acids (PerkinElmer). Complete Mini Protease Inhibitor Cocktail (Roche). Dithiothreitol (Sigma). Protein A agarose (Invitrogen). Cycloheximide (100 µg/ml) (Sigma-Aldrich). MG132 (10 µM, Sigma-Aldrich) and Bafilomycin A1 (400 nM, APExBIO) were used for the designated times. Palmitic acid (400 mM) (Sigma-Aldrich), Oleic acid (400 mM) (Sigma-Aldrich). CellTiter-GloR Luminescent Cell Viability Assay (Promega) was used to measure cell viability according to manual. Lysosomal Intracellular Activity Assay Kit (Abcam) was used to measure lysosomal activity according to user manual. Antibodies for Western blot analysis: Rabbit anti-ATF4 (1:1000, Cell Signaling); Rabbit anti-BiP(GRP78) (1:1000, Cell Signaling); Mouse anti-CHOP (1:1000, Cell Signaling); Mouse anti-eIF2α (1:1000, Santa Cruz); Rabbit anti-eIF2α-phospho (Ser51) (1:1000, Abcam); Rabbit anti-GADD34 (1:1000, Santa Cruz); Rabbit anti-PERK (1:1000, Cell Signaling); Rabbit anti-HERPUD1 (1:2000, Cell Signaling); Mouse anti-α-TUBULIN (1:4000, Sigma-Aldrich); Mouse anti-Proinsulin (1:1000, Novus Biologials); Rabbit anti-MAFA (1:2000, Cell Signaling); Rabbit anti-PCSK2 (1:1000, Cell Signaling); Rabbit anti-PDX1 (1:2000, Cell Signaling); Rabbit anti-NKX2.2 (1:1000, Invitrogen); Guinea pig anti-p62(SQSTM1) (1:1000, Progen Biotechnik); Rabbit anti-LC3(ATG8) (1:1000, Novus biologicals); Guinea pig anti-Insulin antibody (1:200, Covance). Immunofluorescence staining antibodies: Rabbit anti-PDI(P4HB) (1:200, Sigma-Aldrich); Rabbit anti-ERGIC53 (1:200, Sigma-Aldrich); Goat anti-rabbit IgG Alexa-488 (1:200, Invitrogen); Goat anti-mouse IgG Alexa-594 (1:200, Invitrogen); Hoechst33324 (8 µM, Invitrogen).

### Protein sample preparation for Western blot analysis
For protein extraction, cells were washed twice with ice-cooled 1xPBS before lysis. Ice-cooled lysis buffer (50 mM Tris-HCl pH7.5, 150 mM NaCl, 2 mM EDTA, 1% NP-40, 0.1% SDS, 0.5% sodium deoxycholate, supplemented with EDTA-free protease inhibitor (Roche Applied Science) and PhosSTOP phosphatase inhibitor (Roche Applied Science) was added to cells. Cells were scraped off and sonicated on ice. Protein

lysates were centrifuged for 5 min in $10,000 \times g$ at 4 °C. Supernatant was collected and quantified by DC Protein Assay Kit (Bio-Rad). The lysate was diluted to 1 μg/μL by using lysis buffer. The diluted lysates were mixed with 5x sampling buffer (300 mM Tris-HCl pH6.8, 50% glycerol, 10%(v/v) β-mercaptoethanol, 10%(w/v) SDS and 50 mg bromophenol blue) or Pierce Lane Marker Non-Reducing Sample Buffer (Thermo Scientific) in a 1-to-4 ratio for reducing or non-reducing Western blot analysis, respectively. For non-reducing analysis, gels were soaked with 25 mM DTT for 10 min before electrotransfer to Immobilon-P PVDF membrane (Sigma-Aldrich).

### Proinsulin secretion and dot blotting
MIN6 cells were plated onto 10 cm plates and cultured in the cell growth medium until reaching to 60–70% confluency. Cells were treated with 200 μM CPA in a fresh medium until 1 h before the designated treating times. The cell was quickly washed with pre-warmed 1x PBS, before refreshing 10 mL culture medium with or without CPA. Cells were cultured for 1 h at 37 °C. 8 mL culture medium was collected in a 15 mL centrifuge tube and centrifuged at $100 \times g$ for 3 min. Top 4 mL medium of the supernatant was collected for dot blotting analysis. The supernatant was stored in at −80 °C and analyzed within 1 week.

Supernatants containing secreted proinsulin from the treated cells were transferred to PVDF membranes (Thermo Scientific) at increasing amounts (50, 100 and 150 μL) by using Minifold I micro-sample filtration manifold (Schleicher and Schuell). MIN6 cell lysates and fresh medium were used as positive and negative controls, respectively. The membrane was incubated in 1xTBST containing 5% skim milk for 30 min at 25 °C with mild shaking. The membrane was washed twice with 1xTBST, followed by a standard Western blotting protocol.

### Measurement of protein synthesis rate
Protein synthesis rates were measured as previously described[22]. In brief, cells were treated with CPA for the indicated times. At the end of treatments, [³⁵S]Met/Cys (30μCi/mL EXPRE³⁵S Protein Labeling Mix (PerkinElmer)) was added to the cells for an additional 30 min. After labeling, cells were washed and lysed, and the radioactivity incorporated into proteins was determined by liquid scintillation counting. The protein synthesis rate was calculated as the rate of [³⁵S]Met/Cys incorporation to total cellular protein from the same lysate.

### Measuring in vitro guanine nucleotide exchange factor (GEF) activity of eIF2B
EIF2B activity was measured as previously described[22]. In brief, cells were washed and scraped off in homogenization buffer (45 mM HEPES-KOH pH 7.4, 0.375 mM MaOAc, 75 mM EDTA, 95 mM KOAc, 10% glycerol, 1 mM DTT, 2.5 mg/mL digitonin, supplemented with EDTA-free protease inhibitor (Roche Applied Science) and PhosSTOP phosphatase inhibitor (Roche Applied Science)). Cell lysates were homogenized and quantified for protein concentration. The EIF2B activity was calculated as the rate of exchange from eIF2α[³H]GDP to nonradioactive GDP over the time points.

### Metabolic labeling of MIN6 cells and proinsulin immunoprecipitation
MIN6 cells were plated onto 12-well plates and cultured in the cell growth medium. After 0 h, 1 h or 18 h treatment with 200 μM CPA or 0.2% DMSO, cells were quickly washed with Met/Cys-deficient DMEM medium and pulse labeled with ³⁵S-amino acids (Trans³⁵S amino acids) for 15 min at 37 °C. Labeled cells were washed with cold PBS containing 20 mmol/L N-ethylmaleimide (NEM) and lysed in radio-immunoprecipitation assay buffer (25 mmol/L Tris, pH 7.5, 100 nmol/L NaCl, 1% Triton X-100, 0.2% deoxycholic acid, 0.1% SDS, 10 mmol/L EDTA) containing 2 mmol/L NEM and protease inhibitor cocktail.

Equal amount of protein lysates were immunoprecipitated with guinea pig polyclonal anti-insulin antibody and protein A agarose overnight at 4 °C. Immunoprecipitates were washed and analyzed by reducing 4–12% NuPAGE Bis-Tris Gel SDS-PAGE, followed by phosphorimaging, and bands were quantified with ImageJ software.

### Polysome profile and RT-qPCR
MIN6 cells were plated onto 15 cm plates and cultured in the cell growth medium until reaching to 60–70% confluency. After 0 h, 1 h or 18 h treatment with 200 μM CPA, cells were quickly washed with pre-cooled 1x PBS twice, then lysed with RIPA buffer. The sample was layered on 10–60% sucrose gradients (20 mM Tris-HCl ph7.4, 150 mM NaCl, 10 mM MgCl2, 1 mM DTT). The tubes were centrifuged at 36,000 rpm ($221632.5 \times g$) at 4 °C for 3 h. The samples were fractionated into ten collection tubes. The first collection tube was discarded and total RNA of the rest of nine tubes was purified by Trizol (Invitrogen). Reverse transcription was performed with ProtoScript II (New England Biolabs) according to the company instructions. QPCR was done by using the StepOnePlus Real-Time PCR System (Applied Biosystems).

### RNA preparation and qRT-PCR
Cells were washed twice with ice-cooled 1xPBS on ice after the designated treatments. Total RNA was extracted using TRIzol LS reagent (Invitrogen). The cDNA was synthesized using SuperScript III First-Strand Synthesis SuperMix (ThermoFisher). The relative quantity of specific mRNAs was measured by using VeriQuest SYBR Green qPCR Master Mix (ThermoFisher) with StepOnePlus Real-Time PCR System (Applied Biosystem). Primers used in this study are listed in SupplementaryTable 5.

### Immunofluorescence staining
MIN6 and INS1E cells, were plated on glass microscope cover slips (ThermoFisher) in 6 cm culture dishes and were allowed to grow for 48 h. For Human EndoC-βH3 cells, cells were plated on coated (matrigel (1:500), Sigma-Aldrich and fibronectin (1 μg/cm²), Sigma-Aldrich) glass microscope cover slips (ThermoFisher) in 6 cm culture dishes and allowed to grow for 1 week. After the designated treatments, cells were washed twice with ice-cooled 1xPBS on ice. Cells were fixed with 4% paraformaldehyde for 10 min. Fixed cells were washed twice with ice-cooled 1xPBS and incubated in PBST (1xPBS + 0.02% Triton X-100) for 15 min, PBST with 10% FBS for 30 mins, and PBST with 10% FBS and primary antibodies at 4 °C for 16 h. After washing with ice-cooled PBST twice, cells were incubated in PBST with 10% FBS and secondary antibodies for 2 h in dark. This was followed by washing with ice-cooled PBST twice and nuclei staining with Hoechst 33342 for 5 min in dark. After washing with ice-cooled PBST twice, cells were mounted in Fluoromount-G (Electronic Microscope Sciences) and sealed with clear nail polish on microscope slides. The images were captured using a Leica SP8 confocal microscope. Imaging areas were randomly selected in a single-blind manner by a microscope specialist. Four imaging areas were fetched in each condition, while a representative image was shown.

### GSIS in MIN6 cells
MIN6 cells were plated onto 6 cm plates for GSIS analysis. To optimize the measurement, 500,000 cells were plated and medium was refreshed every day. After two days recovery, cells were treated with 200 μM CPA or 0.2% DMSO in a fresh medium of the designated treating times. Cells were quickly washed with pre-warmed KRB buffer (120 mM NaCl, 20 mM HEPES, 5 mM KCl, 2 mM CaCl2, mM MgSO4, 0.5% BSA, pH7.4) twice. After washing, cells were incubated with 1 mL of KRB buffer with 2.8 mM glucose for 1 h at 37 °C. After incubation, the buffer was replaced with 1 mL of KRB with 2.8 mM glucose or 1 mL of KRB with 16.8 mM glucose for 1 h at 37 °C. The supernatant was

collected into a 1.7 mL tube, then centrifuged at $100 \times g$ for 3 min. The upper 1 mL of the supernatant was collected for evaluation of GSIS analysis. The analysis was done by using Mouse Insulin ELISA kit (10-1247-01 Mercodia) according to the manufacturer's instruction.

## Mouse pancreatic islet of Langerhans isolation and GSIS evaluation

Islets were isolated from C57BL/6 J (the Jackson Laboratory, stock no.000664) six-week-old male mice, for ex vivo immuno-fluorescence and GSIS studies. The mouse work was approved and maintained in the Animal Resource Center in Case Western Reserve School of Medicine with all relevant ethical regulations. The mice were maintained on a normal mouse diet and a 12 h/12 h light/dark cycle, with the temperature ranging between 70°F-75°F, the humidity was ranging between 30%-40%. The islet purification protocol has been previously described. Briefly, mice were anes-thetized by Nembutal. The pancreas was removed, briefly washed with 1xPBS twice. The pancreas was digested with 1000 U/mL collagenase for 15 min with shaking (60 rpm). The sample was centrifuged at $290 \times g$ for 30 seconds. The pallet was transferred to a 10 cm plate with pre-warmed islet medium (DMEM supple-mented with10% FBS and 1% PSQ) and cultured at 37 °C for 1 h. Single islets were collected and incubated with pre-warmed fresh islet medium and allowed to recover for 24 h before experi-mentation. For immunofluorescence, 10 islets were collected into a 1.7 mL tube and incubated with 200 µM CPA or 0.2% DMSO for the designated times. Islets were washed with precooled 1xPBS twice and were used for subsequent immunofluorescence experiments. For GSIS, 20 islets were collected into a 1.7 mL tube and incubated with 200 µM CPA or 0.2% DMSO for the designated times. After washing with pre-warmed 1xPBS, islets were incu-bated with 1 mL of KRB buffer with 2.8 mM glucose for 1 h at 37 °C. After incubation, the buffer was replaced with 1 mL of KRB with 2.8 mM glucose or 1 mL of KRB with 16.8 mM glucose for 1 h at 37 °C. The supernatant was collected into a 1.7 mL tube, and then centrifuged at $100 \times g$ for 3 min. The upper 0.5 mL of supernatant was collected for GSIS analysis. The analysis was done by using the Mouse Insulin ELISA kit (10-1247-01 Mercodia) according to the manufacturer's instructions.

## Quantification of MAFA fluorescence intensity in mouse islets

The raw, multiple Z-section confocal images were used for the measurement. To avoid a multiple calculation from the same cell, five z-sections from the upper layers of the islet images were stacked. The fluorescence intensities of MAFA staining and Hoechst 33324 nucleus staining were measured by ImageJ. A related MAFA expression level per cell was calculated by MAFA value/nucleus staining intensity of the same islet cell. The value of MAFA intensity smaller than 50 (8-bit intensity scale) was removed to avoid an inclusion of other cell types in the same islet in the statistical analysis.

## Deep sequencing library generation

The general procedures for generation of mRNA-seq and Ribo-seq libraries were previously reported by us and others. In brief, cells were washed twice by ice-cooled PBS with 100 µg/ml cycloheximide (Sigma-Aldrich) before harvest. Lysis buffer (10 mM Tris, pH 7.4, 100 mM NaCl, 140 mM KCl, 10 mM MgCl₂, 1 mM DTT, 1% (v/v) Triton X-100, 500 U/mL RNasin and 100 µg/mL cycloheximide) was added and cells were scraped fast on ice. The lysate was centrifuged for 5 min in $500 \times g$ at 4 °C. The supernatant was collected and centrifuged for 10 minutes in $10,000 \times g$ at 4 °C. The supernatant was collected for mRNA and monosome enrichments. For mRNA enrichment, total RNA was extracted from lysates with TRIzol LS reagent (Invitrogen). The RNA with poly adenosine tract was purified by a Magnetic mRNA Isolation

Kit (New England Biolabs) according to the user manual. The RNA with poly adenosine tract was fragmented with an alkaline buffer (2 mM EDTA, 10 mM Na₂CO₃, 90 mM NaHCO₃, pH ≈ 9.3) treatment for 40 min at 95 °C. For monosome enrichment, lysates were treated with RNase If (50U/100 µg, New England Biolabs) for 40 mins at 25 °C. Ice-cooled open top 10–50% sucrose gradient centrifuge tubes (14 × 89 mm, SETON) were generated by Gradient master 108 (Biocomp). The digested lysate was loaded on top of the gradients and were cen-trifuged in a SW41Ti rotor (Beckman) for 2 h in 40000 rpm (273620.3 × g) at 4 °C in an L-70 ultracentrifuge (Beckman). The monosome enriched fraction was collected by a PGFip Psiton Gradient Fractionator (Biocomp). Total RNA was purified and extracted by TRIzol LS reagent. RNA length between 20 to 40 bases was size selected by 15% TBU gel (Invitrogen). The 3′ ends of the selected RNA fragments were dephosphorylated by T4 PNK (New England Biolabs). Ribosomal RNA was removed from the RNA isolated from monosome enrichment, by NEBnext rRNA depletion kit (New England Biolabs). RNA was ligated with Linker-A (New England Biolabs) by T4 RNA ligase II KQ (New England Biolabs). Ligated RNA fragments were size selected by 10% TBU gels (Invitrogen). Selected RNA was reverse transcribed (RT) with in-house-designed DNA oligos (Supplementary Data 5) by ReverTra Ace (Toyobo) according to the user manual. The RT product was circularized with CircLigase II ssDNA Ligase (Lucigen). The librar-ies were amplified by Phusion High-Fidelity DNA Polymerase (New England Biolabs) for 10 polymerase-chain-reaction (PCR) cycles with in-house-designed indexed PCR oligos (Supplementary Data 5). PCR products between 140 to 170 bps were collected by 4–20% TBE gel (Invitrogen) for deep sequencing. The libraries were quantified and sequenced by Novogene Corporation in Illumina HiSeq platform. Three independent experiments were performed for bioinformatics analysis.

## Islet microdissection and microarray analysis

The existing microarray raw data was utilized in this study; the pre-paration was described previously[30,31]. In brief, frozen tissue was obtained from the Network for Pancreatic Organ donors with Diabetes (nPOD) (Supplementary Data 4)[87]. Optimal cutting temperature slides of pancreatic tissue were used for laser-capture microscopy that was conducted on an Arcturus Pixcell II laser capture microdissection system (Arcturus Bioscience). All islets visible in two to five sections from each sample were pooled, and RNA was extracted using the Arcturus PicoPure RNA Isolation Kit (Applied Biosystems). RNA quan-tity and quality were determined using a Bioanalyzer 2100 (Agilent Technologies). RNA samples were subjected to gene expression ana-lysis using Affymetrix expression arrays (ThermoFisher). Fluorescence intensity was used to refer to gene expression. Data was aggregated to minimize the identifiable information before analysis.

## Single-cell RNA sequencing analysis

The scRNA-seq raw data from T1D and healthy donors datasets were obtained from HPAP (https://hpap.pmacs.upenn.edu/)[61]. The raw data were pooled and normalization using SCtransform[88] in the Seurat package in R. No additional batch-effect removal step was applied. The normalized dataset contains gene expression profiles of the 35041 islet cells, along with 23509 gene expression levels. Cell types were deter-mined by Seurat clustering results according to the expression of pancreatic marker genes[89]. In brief, we performed a non-supervised Uniform Manifold Approximation and Projection (UMAP)[90] of the dataset to cluster cells with similar expression profiles. Totally 8349 cells, which contain β-cell identities, were defined as β-cell for the study. For the UPR study, the cell profiling was done by a machine learning approach via the Rtsne package in R according to the expression of the UPR signature (GO:0030968) and 6 β-cell identity genes, *INS*, *PAX6*, *NKX2.2*, *NKX6.1*, *MAFA* and *PDX1*. Patients' informa-tion used in this study was listed in Supplementary Fig. 4b.

## in silico pseudo-time reconstruction of scRNA-seq

The pseudo-time reconstruction of scRNA-seq was done by Monocle 3[67] and TSCAN[66] packages in R in a default setting. In brief, a dataset contained the expression information of the UPR signature (GO:0030968) and 6 β-cell identity genes of the 8349 cells. Data were preprocessed to reduce dimensionality, via UMAP for single-cell trajectory. The expressions of the average of β-cell identity genes and BiP along the trajectory were fetched for the analysis.

## Cholesterol synthesis measured by GC-MS

Cholesterol synthesis was determined following deuterium labeling incorporation from deuterium-enriched media. In brief, MIN6 cells were incubated in the medium containing 10% deuterated water ($^2H_2O$, molar percent enrichment) with CPA for 18 h (CPA18); while without CPA served as control. For the recovery sample, after 18 h of CPA treatment, the cells were washed twice with 1xPBS and refreshed with the medium containing 10% deuterated water for 24 h incubation. After incubation, the supernatant was collected for determination of deuterium enrichment in the medium. After 1xPBS wash twice, cells were trypsinized and pelleted by centrifugation at 4 °C at $650 \times g$ for 5 min. Cell pellets were washed with 1xPBS and pelleted by centrifugation at 4 °C at $650 \times g$ for 5 min. Supernatants were discarded and pre-cooled methanol was added into the tubes containing cell pellets. After mixing by inversion for 5 times, cells were pelleted by centrifugation at 4 °C at $650 \times g$ for 5 min. The supernatant was discarded and pellets were stored at −80 °C until extraction.

The $^2H$-labeling of body water was determined by the exchange with acetone, as previously described by[91] and further modified. In brief, 5 μL of cellular medium or standard were reacted for 4 h at room temperature with 5 μL of 10 N KOH and 5 μL of acetone. Headspace samples were injected into GC/MS in split mode (1:40). Acetone m/z 58 and 59 were monitored. Isotopic enrichment was determined as ratio of m/z 59/(58 + 59) and corrected using a standard curve.

For cholesterol extraction, cell pellets in the tubes were homogenized frozen in 600 μl of Folch solution (chloroform:methanol, 2:1, v/v) on dry ice. After addition of 0.4 volumes of ice-cold water, cells were homogenized again and let incubate on ice for 30 min. Homogenates were centrifuged at 4 °C at $20,000 \times g$ for 10 min. Upper methanol/water layer was discarded. Internal standard (heptadecanoic acid) was added to the bottom chloroform layer and was evaporated to dryness. Cholesterol was then converted to TMS (trimethylsilyl) derivative by reacting with bis(trimethylsilyl) trifluoroacetamide with 10% trimethylchlorosilane (Regisil) at 60 °C for 20 min. Resulting TMS derivatives were run in GC-MS.

GC-MS conditions used for the analysis were carried out on an Agilent 5973 mass spectrometer equipped with 6890 Gas Chromatograph. A DB17-MS capillary column (30 m × 0.25 mm × 0.25 μm) was used in all assays with a helium flow of 1 mL/min. Samples were analyzed in Selected Ion Monitoring (SIM) mode using electron impact ionization (EI). Ion dwell time was set to 10 msecs. Cholesterol m/z 368−372 were monitored to detect isotopic label incorporation.

The absolute cholesterol biosynthesis was computed as follows. Cholesterol GC-MS data were corrected for natural abundance and resulting enrichments were divided into precursor isotopic enrichment of media deuterium (10%) to follow precursor-product relationship. This resulted in fractional synthesis rate. Absolute cholesterol synthesis rate was determined by multiplying fractional rate by the cholesterol content which was determined using internal standards.

## Biostatistics and bioinformatics

Mouse RNA reference (GRCm38.p6.rna) was used for sequencing reads alignment. The alignment was done by Bowtie2[92] in an in-house-established Galaxy platform. In brief, raw sequencing reads were filtered by quality, Q30 > 90%. The length of reads between 15 and 40 were selected. The selected reads were aligned to the RNA reference with parameters of equal or higher than 95% identity and less than one mismatch was allowed. Genes with at least one read aligned and identified in all three independent experiments were collected for further analysis. Transcript count was calculated as reads per kilo-base per million reads of total aligned reads (RPKM). Ribosome occupancy (Ribo$^{ocp}$) of a transcript was calculated as the mean of transcript count in Ribo-seq divided by the mean of transcript count in mRNA-seq. Gene functional annotations were done in KEGG: Kyoto Encyclopedia of Genes and Genomes (https://www.genome.jp/kegg/) and Gene Ontology Resource (http://geneontology.org/). Data analysis, statistics and presentations were done by R (https://www.r-project.org/) and Excel (Microsoft). Bar charts were done by Excel. Heatmaps were done by Heatmap2 package in R. Scatterplots, volcano plots and violin plots were done by the Ggplot2 package in R.

The statistics of the heatmaps in Fig. 3 was done by Excel. For Fig. 3c, the changes of transcript expression in [log2] values of CPA1 over control (acute), CPA18 over CPA1 (chronic) and CPA18 over control (overall) were calculated. The cutoffs to define whether the transcript was upregulated or downregulated were set as 0.32 or −0.32, respectively. Transcripts that showed upregulation in both acute and chronic phases were grouped into G1. Transcripts that showed upregulation in the acute phase but downregulation in the chronic phase were grouped into G2. Transcripts that showed no change in the acute phase but upregulation in the chronic phase but overall upregulation were grouped into G3. Transcripts that showed downregulation in both, acute and chronic phases were grouped into G4. Transcripts that showed downregulation in the acute phase but upregulation in the chronic phase were grouped into G5. Transcripts that showed no change in the acute phase but downregulation in the chronic phase but overall downregulation were grouped into G6. For Fig. 3d, transcripts were grouped separately according to the Ribo$^{ocp}$ values. Transcripts that showed increased Ribo$^{ocp}$ in both acute and chronic phases were grouped into G7. Transcripts that showed increased Ribo$^{ocp}$ in the acute but decreased Ribo$^{ocp}$ in the chronic phase were grouped into G8. Transcripts that showed no change of Ribo$^{ocp}$ in the acute phase and an increase in the chronic phase, leading to overall increased Ribo$^{ocp}$ were grouped into G9. Transcripts that showed no change of Ribo$^{ocp}$ in the acute phase and a decrease in the chronic phase, leading to overall to decreased Ribo$^{ocp}$ were grouped into G10. Transcripts that showed decreased Ribo$^{ocp}$ in the acute, but increased Ribo$^{ocp}$ i in the chronic phase were grouped into G11. Transcripts that showed decreased Ribo$^{ocp}$ in both acute and chronic phases were grouped into G12. The functional enrichments of the 12 groups were analyzed by the KEGG pathway via the DAVID bioinformatics database. The number of genes identified in a single biological pathway was denoted.

For the transcriptome and translatome analysis the gene expression levels represent the means of RPKM of three independent experiments. The values of differential expression were determined as log2(expression in RPKM in condition 2 – expression in RPKM in condition 1) (L2DE). The cuff-off value for data filtering and statistics was set as log2 = 0.6 or −0.6, which represented at least 50% difference in RPKMs between two conditions. The definition of expression regulation was set as described previously (Guan et al. 2017). In brief, up-regulation in both mRNA abundance and ribo$^{ocp}$, $-0.6 \leq (x-y) < 0.6$, $x \geq 0.6$, $y \geq 0.6$; up-regulation in ribo$^{ocp}$ only, $(x-y) < -0.6$, $y > 0.6$; down-regulation in both mRNA abundance and ribo$^{ocp}$, $-0.6 \leq (x-y) < 0.6$, $x \leq -0.6$, $y \leq -0.6$; down-regulation in ribo$^{ocp}$ only, $(x-y) > 0.6$, $y < -0.6$, where x represents the L2DE in transcriptomic data and y represents the L2DE in ribosome footprints.

## Reporting summary

Further information on research design is available in the Nature Research Reporting Summary linked to this article.

## Data availability

The RNA-seq and Ribo-seq sequencing data of the CPA treated MIN6 cells and the control generated in this study are available in the Gene Expression Omnibus (GEO) database under accession code GSE174679. All data and resources supporting the findings described in this manuscript are available in the article and in the Supplementary Information and from the corresponding author upon reasonable request. Source data are provided with this paper.

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

## Acknowledgements

This manuscript used raw data acquired from nPOD (www.jdrfnpod.org) and data acquired from the Human Pancreas Analysis Program (HPAP-RRID:SCR_016202) Database (https://hpap.pmacs.upenn.edu), a Human Islet Research Network (RRID:SCR_014393) consortium (UC4-DK-112217, U01-DK-123594, UC4-DK-112232, and U01-DK-123716). Raw data was aggregated to minimize the identifiable information before analysis. Funding: DK053307 and DK060596 (to M.H.), DK48280, DK111174 and DK127747 (to P.A.), DK130919 and R56 DK128136 (to F.E.). DK112217 and DK123594 (to K.H.K.). JDRF and Hensley Trust, and UC4DK104155 (to C.M.). H.L. is supported by a University of Wisconsin Stem Cell and Regenerative Medicine Center Graduate Fellowship.

## Author contributions

C.-W.C., M.H. and F.E. provided the conceptual framework for the study. C-W.C. designed the experiments. C.-W.C., B.-J. G., M.R.A., Z.G., S.B. and I.B. performed the experiments and collected the data. C.-W.C., H.L. and T.L. performed bioinformatics analysis of MIN6 cells and MEFs. C.-W.C., C.E.M. and I.C.G. performed microarray analysis. C.-W.C., L.G. and K.H.K. performed scRNA-seq analysis and assisted with evaluation of the data. R.J. and C.A.M. performed the polysome profile analysis. J.W. and A.H.Z. performed the experiments with mouse islets. C.-W.C., L.H., A.E.S., P.A., B.T., F.E. and M.H. interpreted the results. M.H. and F.E supervised the project with input from C.-W.C. and B.T. M.H., F.E. and B.T. wrote the paper with input from all authors.

## Competing interests

The authors declare no competing interests.
