## [Peer Review File · Nature Communications]

Title: Adaptation to chronic ER stress enforces pancreatic β -cell plasticityREVIEWER COMMENTS

Reviewer #1 (Remarks to the Author):

This study focuses on specific adaptation of pancreatic beta cells to chronic ER stress. In the study, MIN6 cells were treated with a reversible chemical (CPA) for different times to induce ER stress in acute and chronic scenarios. By means of bio-informational research, the distinct transcriptional and translational reprogramming of beta cells was revealed, indicating the plasticity of beta cells in response to chronic ER stress. A set of master regulators were found, which was collectively termed as “adapto-some”, including genes related to ribosome, protein processing, degradation, et al. Beta cells adapted to chronic ER stress with the change of these regulators, and regained their identity when relieved from stress. After several frequent repeated cycles of ER stress, the plasticity of beta cells was lost gradually. The study also used single cell RNA-seq analysis of islets from type 1 diabetes (T1D) patients to further clarify the relationship of deregulation of the stress-adaptation program and the progression of the T1D. Based on the results, it introduces the hypothesis that beta cell adaptive exhaustion is a component of the pathogenesis of T1D. The paper contains 7 figures and 7 supplementary figures, comprehensively studies the transcriptional and translational reprogramming of beta cells in chronic ER stress. In general, the manuscript was well written, and the experiments were done carefully with appropriate controls. Several points need to be clarified:

Major points:

- 1. In Figure 1 B&E, protein synthesis rate was examined. Did you measure the specific synthesis rate of proinsulin?
- 2. In Figure 3E, several genes related to degradation were revealed. Figure 5 further detected the autophagosome-to-lysosomal degradation pathway. Did you compare the degradation of proinsulin/insulin in chronic stress and after the stress was removed?
- 3. As quoted in the manuscript, the impaired proinsulin processing and secretion are hallmarks of T1D and T2D. The study examined the proinsulin level in steady in the presence of CPA and after CPA was washed out. Did you measure the secretion of proinsulin? Meanwhile, is glucose-induced insulin secretion exhibited significant difference in chronic stress and after the stress was removed?

Minor points:

- 1. In this study, MIN6 cells exhibited adaptation to CPA-induced chronic ER stress and regained their identity when relieved from stress. Frequent repeated episodes of ER stress led to the decrease of cell plasticity. Did you test other ER stress inducers? Is this CPA specific or general?
- 2. According to Figure 7C&D, frequent episodes of chronic ER stress led to decreased plasticity and loss of beta cell specific gene expression, which is impressive and promising. I suggest enrolling more data based on this experiment in the future. It might be more helpful to discover the temporal changes of the regulators in the progression of diabetes.

Reviewer #2 (Remarks to the Author):

This study explored whether duration and frequency of chronic ER stress directly compromised β -cell identity. Chronic ER stress caused β -cells to undergo transcriptional and translational reprogramming which was reversible. Expression of regulators of β -cell function and identity and proinsulin processing were impaired but after recovery from stress cells regained their identity. A threshold of stress episodes was found after which β -cell identity progressively declined. Single cell RNA-seq of islets from T1D patients showed deregulation of the chronic stress-related genes.

While there are multiple studies on metabolic ER stress on β -cell function this paper takes the novel approach of investigating β -cell resilience and the ability to recover from repeated episodes of stress at transcriptomic and proteomic level. As such represents novel findings. The majority of the data is definitive and accurately interpreted. A few issues are raised below.

A limitation on interpretation from the study is the use of the Min6 insulinoma cell line only and a single inducer of ER stress, CPA. While this minimal model provides proof of principle it will not reproduce the range of stressors in human diabetes that are likely to include hyperglycemia, hyperlipidemia, cytokine action etc. Also, the homeostatic control within islets due to the minute-by-minute paracrine interactions of insulin, glucagon, somatostatin, 5-HT etc. cannot be mimicked. This could be put in context in the Discussion along with possible approaches to better represent real pathophysiology.

The changes to gene expression and cellular stress pathways can partly be addressed from the scRNAseq study where relative gene expression changes for alpha, delta and PP cells for T1D and T2D have been measured. Other than for beta cells the transcriptomes for the other endocrine cell types have scant mention. Are delta cell ER stress pathways equally affected?

Please give the source of the Min6 cells. A number of variant strains now exist in different labs with repeated passage and it is valuable to relate to other papers that used the same source.

In Figure 1C a partial recovery of eIF2B is shown with washout after 12 h, but compared to control is it statistically any different at any time between 1 and 12h? If not then the recovery had occurred for all of these time points.

Figure 1I – Is it possible to quantify the proinsulin puncta as in Fig. 2 since this would be valuable quantitative data?

Reviewer #3 (Remarks to the Author):

The study by Chen et al relates to a relevant question for both type 1 and type 2 diabetes field, as ER

stress is thought to be causative of beta-cell dysfunction in early developments of both conditions. However the findings reported are mostly corroborative of known mechanisms of beta-cell adaption to ER stress, and do not represent a significant advance as compared to the current knowledge in the field.

Major points:

1. Lack of clear major novelty point or concept: it is already widely accepted that beta-cell adapt to ER stress by blocking mRNA translation initiation while simultaneously promoting the translation initiation of protective mechanisms (chaperones..). When this response is unable to resolve ER stress, the beta-cells become dysfunctional and/or die. There may also be an overuse of novel terminology (β EAR, transcriptional and translational reprogramming, beta-cell specific adaptosome, regulatome...) that makes it difficult for the reader to determine what is really relevant and novel.

2. The experimental model used (proliferative murine insulinoma cell line MIN6 subjected to a chemical treatment) is not very innovative as many studies have reported the effect of drug-induced ER stress on such model (MIN6 or INS1 with thapsigargin or tunicamycin treatment). Also the claims of 'chronic' vs 'acute' effects based on 18h and 2h treatment time is not justified. The authors should explain the choice for this drug/dose/timing. It's also crucial to describe the effect of the two CPA treatments on proliferation and cell death as these may influence the interpretation of some of the 'recovery' data presented throughout the manuscript. For instance, in Supp Fig 5C starting point is not shown, how much cell death there is post CPA6h,18, and 42h treatment? It's also important to determine if beta-cell function (as assessed by glucose-stimulated insulin secretion) is altered in this model (or after repeated cycles of CPA treatment).

3. The strategy chosen of comparing CPA-treated MIN6 and MEFs is unclear and potentially misleading as the difference between the 2 lines is much more than only secretory vs non-secretory cells (primary vs transformed cells; embryonic vs adult; etc...). Also MEFs are often used as feeder cells given their capacity to secrete growth factors. Furthermore, conceptually it's not certain that the cause of ER stress-induced beta-cell failure are beta-cell specific genes.

4. One can acknowledge the effort to relate the findings to human primary cells by using available datasets, however single-cell transcriptomics performed on human islet cells (that are not proliferative) subjected to the same CPA treatments would have been of great interest, in order to rule out transformed (proliferative) murine cell line-specific effects.

5. The scRNAseq dataset of islets from T1D and non-diabetic individuals appears as one of the main novelty points of the study. However there are no information given on the characterization of the dataset (quality controls etc..) and the paper referred to (Kaestner et al 2019) is only a description of the HPAP consortium. It would be informative for the reader of this manuscript to know how the donors were selected as beta-cells were found in all 6 T1D donors included and at a rather normal frequency in most of the cases. A number of additional analyses can be performed such as pseudo timing that may reveal different degrees of maturity of the beta-cells. On Fig 7B, there is no annotation of which cells belong to which donor, and which category of donors (ND, T1D).

6. The experimental set-up of ER stress cycles is potentially interesting but it's critical to know how much cell loss (death) there is per cycle. Some information on cell numbers (reflecting both proliferation and death) is needed. Along this line, cleaved Caspase 3 could be checked by WB at the different time points. On Fig 7C, please show all data, also for cycles 2-5. How is the expression of the classical ER stress markers ATF3 and XBP1s/u (qPCR) and of insulin at protein level (WB)?

Other points:

1. One could expect that the loss of beta-cell maturity markers is associated with an increase in genes that are normally not expressed in adult beta-cells, such as progenitor markers (dedifferentiation) or an increase in markers specific for other islet cell types (transdifferentiation), as reported in a number of beta-cell dedifferentiation studies both in murine and human beta-cells (though the markers are different). Is that the case in these experimental models with (cycles of) CPA treatment?

2. Fig 1F: why is BIP already highly expressed in the control condition?

3. Fig 2: Claims made by the authors regarding the alteration in the processing of proinsulin (now done exclusively by fluorescence imaging) would be strengthened by a more standard biochemical approach (pulse-chase).

4. Although the Discussion section is rather long, there is no discussion of the limitations of the study.

5. There is no quantification of the effect described on Fig 1I.

6. Supp Table 1 is missing.

7. Gene lists corresponding to all main figures should be added as supplementary information.

8. Please justify how the 49 UPR-related genes have been defined.

Response to reviewers

Reviewer #1 (Remarks to the Author):

This study focuses on specific adaptation of pancreatic beta cells to chronic ER stress. In the study, MIN6 cells were treated with a reversible chemical (CPA) for different times to induce ER stress in acute and chronic scenarios. By means of bio-informational research, the distinct transcriptional and translational reprogramming of beta cells was revealed, indicating the plasticity of beta cells in response to chronic ER stress. A set of master regulators were found, which was collectively termed as “adaptosome”, including genes related to ribosome, protein processing, degradation, et al. Beta cells adapted to chronic ER stress with the change of these regulators, and regained their identity when relieved from stress. After several frequent repeated cycles of ER stress, the plasticity of beta cells was lost gradually. The study also used single cell RNA-seq analysis of islets from type 1 diabetes (T1D) patients to further clarify the relationship of deregulation of the stress-adaptation program and the progression of the T1D. Based on the results, it introduces the hypothesis that beta cell adaptive exhaustion is a component of the pathogenesis of T1D. The paper contains 7 figures and 7 supplementary figures, comprehensively studies the transcriptional and translational reprogramming of beta cells in chronic ER stress. In general, the manuscript was well written, and the experiments were done carefully with appropriate controls. Several points need to be clarified:

We thank the reviewer for the positive comments about our manuscript.

Major points:

- 1. In Figure 1 B&E, protein synthesis rate was examined. Did you measure the specific synthesis rate of proinsulin?

In response to the reviewer's request we have used two approaches to demonstrate the regulation of proinsulin synthesis during progression of β -cells from acute to chronic ER stress. The first approach was metabolic labeling of proinsulin in MIN6 cells with a 15 min pulse in media containing $^{35}\text{S}/\text{Cys}/\text{Met}$ at 1h or 18h of treatment with CPA. Labelled proinsulin was immunoprecipitated and analyzed on an SDS-PAGE gel. It is shown in two biological replicates that synthesis of proinsulin was repressed at 1h and de-repressed at 18h (**Sup. Fig. 1C**, in the revised manuscript and below). The second approach was to demonstrate that the efficiency of translation of the proinsulin mRNA also decreased at 1h and was recovered at 18h of treatment with CPA. We showed this via evaluation of proinsulin mRNA distribution on polysome profiles at the different times of treatment with CPA. It is shown that association of proinsulin mRNA with polyribosomes decreased from 89% in control untreated cells to 34% during acute stress and returned to 68% during chronic stress (**Sup. Fig. 1D**, in the revised manuscript and below). As expected translation of the ATF4 mRNA was activated at 1h of CPA-treatment and remained activated during chronic ER stress. Taken together,

these data suggest that regulation of proinsulin mRNA translation follows the regulation of global protein synthesis rates during adaptation to ER stress.

The following text was added in the revised manuscript (page 6):

To demonstrate the regulation of proinsulin synthesis during progression of β -cells from acute to chronic ER stress, we followed two approaches. The first approach was metabolic labeling of proinsulin in MIN6 cells with a 15 min pulse in media containing ^{35}S /Cys/Met at 1h or 18h of treatment with CPA. Labeled proinsulin was immunoprecipitated and analyzed on an SDS-PAGE gel. It is shown in two biological replicates that synthesis of proinsulin was repressed at 1h and de-repressed at 18h (**Sup. Fig. 1C**). The second approach was to demonstrate that the efficiency of translation of the proinsulin mRNA also decreased at 1h and was recovered at 18h of treatment with CPA. We showed this via evaluation of proinsulin mRNA distribution on polysome profiles at different times of CPA treatment. It is shown that association of proinsulin mRNA with polyribosomes decreased from 89% in control untreated cells to 34% during acute stress and returned to 68% during chronic stress (**Sup. Fig. 1D**). As expected, translation of the ATF4 mRNA was activated at 1h of CPA treatment and remained activated during chronic ER stress. In contrast, translation of the ribosomal protein RpL13A mRNA was severely inhibited during acute stress and only partially recovered during chronic stress (**Sup. Fig. 1D**).

Supplementary Figure 1.

-2. In Figure 3E, several genes related to degradation were revealed. Figure 5 further detected the autophagosome-to-lysosomal degradation pathway. Did you compare the degradation of proinsulin/insulin in chronic stress and after the stress was removed?

In response to the reviewer's request, we compared the degradation of proinsulin in control non-stressed MIN6 cells and cells exposed to chronic ER stress via the use of the protein synthesis inhibitor, cycloheximide (CHX). As expected, we found that proinsulin in control cells is very unstable as previously reported. In contrast, proinsulin was very stable during chronic ER stress (**Sup Fig. 1E** in the revised manuscript and below), in agreement with the decreased lysosomal activity (**Fig. 5D**). Furthermore, proinsulin degradation occurred via lysosomes (**Fig. 5G** of the revised manuscript) and not the proteasome (**Sup. Fig. 1E** in the revised manuscript and below). Inhibition of the proteosomal activity with MG132, had no effect on proinsulin accumulation during chronic ER stress. Other stress-induced proteins were targeted for degradation by the proteasome, confirming the function of MG132 in proteosomal activity inhibition (**Sup. Fig. 1E** in the revised manuscript and below). Taken together, the resilience of β -cells to chronic ER stress, includes positive and negative mechanisms of regulation of proinsulin gene expression. Despite decreased proinsulin mRNA levels, proinsulin synthesis and stability are enhanced, thus preparing the cells to recover when stress is removed. The washout of the stress, returned proinsulin regulation to control conditions; increased mRNA levels (**Fig. 1G** in the revised manuscript) and short half life (data not shown).

The following text has been added in the revised manuscript (page7):

We further tested this hypothesis by comparing the degradation of proinsulin in control non-stressed MIN6 cells and cells exposed to chronic ER stress via the use of the protein synthesis inhibitor, cycloheximide (CHX). We found that proinsulin in control cells is very unstable as previously reported (Liu et al., 2018). In contrast, proinsulin was very stable during chronic ER stress. (**Sup. Fig. 1E**). Furthermore, proinsulin degradation was insensitive to inhibition of the proteasome. Inhibition of the proteosomal activity with MG132, had no effect on proinsulin accumulation during chronic ER stress (**Sup. Fig. 1E**). Other stress-induced proteins were targeted for degradation by the proteasome, confirming the effect of MG132 in inhibiting the proteosomal activity (**Sup. Fig. 1E**). Finally, functional recovery of CPA-treated MIN6 cells during CPA washout conditions, indicated partial recovery of the CPA-inhibited glucose stimulated insulin secretion (GSIS) (**Sup. Fig. 1F**). Taken together, the resilience of β -cells to chronic ER stress, includes positive and negative mechanisms of regulation of proinsulin gene expression. Despite decreased proinsulin mRNA levels, proinsulin synthesis and stability are enhanced, thus preparing the cells to recover when stress is removed.

Supplementary Figure 1.

-3. As quoted in the manuscript, the impaired proinsulin processing and secretion are hallmarks of T1D and T2D. The study examined the proinsulin level in steady in the presence of CPA and after CPA was washed out. Did you measure the secretion of proinsulin?

In response to the reviewer's question, we measured secretion of proinsulin in MIN6 cells treated with CPA for 1h or 18h, followed with washout of CPA after 18h, for either 1h or 6h. We evaluated the levels of proinsulin in the media of the treated cells via dot blot analysis. In all conditions, secretion of proinsulin was detected in media following 1h incubation with MIN6 cells. We also used MIN6 cells (not-treated with CPA) as a negative control and cell extracts as a positive control. Densitometric analysis of the dot blot was used to quantify proinsulin secretion (**Sup. Fig. 2A,B** in the revised manuscript and below). We found that proinsulin is secreted during treatment with CPA and secretion decreases during washout of CPA from cells. This is in line with multiple studies indicating the increased secretion of proinsulin from stressed β -cells (PMID: 27541297).

The following text has been added in the revised manuscript (page 8):

Because plasma proinsulin in the blood is considered a risk factor in diabetes (Reaven et al., 1993), we also determined if proinsulin is secreted from MIN6 cells treated with CPA for 1h and 18h or washout of CPA, following CPA treatment for 18h. We evaluated the levels of proinsulin in the media of the treated cells via dot blot analysis (ref). In all conditions, secretion of proinsulin was measured in media added to the cells for 1h. We also used non-treated with CPA cells as negative control and cell extracts as a positive control for the dot blot analysis. Densitometric analysis of the dot blot was used for quantification of the secreted proinsulin (**Sup. Fig. 2A,B**). It is shown that proinsulin is secreted during treatment with CPA and secretion decreases during washout of CPA from cells. This is in line with multiple studies indicating the increased secretion of

proinsulin from stressed β -cells (Asari et al., 2017). These data support the idea that ER stress-mediated accumulation of proinsulin in β -cells can lead to its secretion to the extracellular space.

Supplementary Figure 2.

A.

B.

Meanwhile, is glucose-induced insulin secretion exhibited significant difference in chronic stress and after the stress was removed?

In response to the reviewer's request, we evaluated GSIS in MIN6 cells in control (non-stressed), in chronic ER stress (18h treatment with CPA) and after washout of stress (18h CPA treatment followed by 24h washout) conditions. We show that in response to chronic ER stress cells blunted GSIS, but GSIS was restored upon washout from stress (**Sup. Fig. 1F** in the revised manuscript and below).

The following text has been added in the revised manuscript (page 7):

Finally, functional recovery of CPA-treated MIN6 cells during CPA washout conditions, indicated partial recovery of the CPA-inhibited glucose stimulated insulin secretion (GSIS) (**Sup. Fig. 1F**).

Supplementary Figure 1.

Minor points:

- 1. In this study, MIN6 cells exhibited adaptation to CPA-induced chronic ER stress and regained their identity when relieved from stress. Frequent repeated episodes of ER stress led to the decrease of cell plasticity. Did you test other ER stress inducers? Is this CPA specific or general?

In response to the reviewer's request, we tested whether treatment with palmitic acid elicits a similar response in β -cell marker suppression in MIN6 cells to that of CPA. We evaluated levels of proinsulin and levels of the transcription factor MAFA. We show that palmitic acid induces ER stress (**Sup. Fig. 8B**, in the revised manuscript) and decreases proinsulin and MAFA levels (**Sup. Fig. 8C** in the revised manuscript and below). Washout of palmitic acid, restored levels of proinsulin and MAFA, and as expected, stress-induced proteins were undetectable. In conclusion, physiological ER stressors, such as saturated fatty acids, show similar reversibility of stress-induced decline of β -cell specific gene expression. More detailed studies will be necessary to establish the unique stress response elements of β -cells to fatty acids as compared to the bona fide ER stressor, CPA.

The following text was added in the revised manuscript (page 25):

Although our current study modeled changes typical of pre-T1D, alternative experimental systems can study T2D pathogenesis (Swisa et al., 2017). For example, lipotoxicity can induce ER stress in tissue cultured β -cells (Cunha et al., 2008), and is a factor associated with β -cell loss in T2D (Cnop et al., 2012). Similar to the studies presented here via the use of a chemical ER stressor, palmitic acid-induced ER stress in MIN6 cells, results in a similar reversible decrease in MAFA protein levels (**Sup. Fig. 8B,C**). Future studies can determine the commonalities between different physiological and pathological stressors and the bona fide ER stress cellular response described here.

Supplementary Figure 8.

C.

-2. According to Figure 7C&D, frequent episodes of chronic ER stress led to decreased plasticity and loss of beta cell specific gene expression, which is impressive and promising. I suggest enrolling more data based on this experiment in the future. It might be more helpful to discover the temporal changes of the regulators in the progression of diabetes.

We thank the reviewer for this comment. We do aim to investigate temporal regulators of this process in future studies. We plan to perform ribosome profiling after cycles of adaptive exhaustion in order to identify markers for progression of diabetes. We also want to share with the reviewer another exciting preliminary finding. We have begun investigating the mechanism of stress-induced decreases in mRNA levels for a limited set β -cell identity genes: proinsulin, MAFA, PDX1, NKX2-2 and PAX6; and found that the decreased levels in their mRNAs is not due to reduced half lives. We predict another level of regulation during resilience of β -cells to ER stress is the epigenome. We are very excited to investigate this line of research as well, and wish to share our thoughts with the reviewer.

Reviewer #2 (Remarks to the Author):

This study explored whether duration and frequency of chronic ER stress directly compromised β -cell identity. Chronic ER stress caused β -cells to undergo transcriptional and translational reprogramming which was reversible. Expression of regulators of β -cell function and identity and proinsulin processing were impaired but after recovery from stress cells regained their identity. A threshold of stress episodes was found after which β -cell identity progressively declined. Single cell RNA-seq of islets from T1D patients showed deregulation of the chronic stress-related genes.

While there are multiple studies on metabolic ER stress on β -cell function this paper takes the novel approach of investigating β -cell resilience and the ability to recover from repeated episodes of stress at transcriptomic and proteomic level. As such represents novel findings. The majority of the data is definitive and accurately interpreted.

We thank the reviewer for the positive comments and the suggestion of the term resilience. We find this term very appropriate for our study and we adopted it in the revised manuscript. Once again, we thank the reviewer for the suggestion.

A few issues are raised below. A limitation on interpretation from the study is the use of the Min6 insulinoma cell line only and a single inducer of ER stress, CPA. While this minimal model provides proof of principle it will not reproduce the range of stressors in human diabetes that are likely to include hyperglycemia, hyperlipidemia, cytokine action etc. Also, the homeostatic control within islets due to the minute-by-minute paracrine interactions of insulin, glucagon, somatostatin, 5-HT etc. cannot be mimicked. This could be put in context in the Discussion along with possible approaches to better represent real pathophysiology.

We agree with the reviewer that mimicking the range of stressors in human diabetes that include hyperglycemia, hyperlipidemia, cytokine action etc. or paracrine interactions in the islets environment is not possible with our experiments and it is a limitation of our study. However, as the reviewer also acknowledges, MIN6 cells are a good system to start discovering factors involved in the resilience of β -cells to chronic ER stress. We envision that similar studies to the ones performed in our manuscript can test the effects of cytokines, lipids, glucose concentrations, etc; and this approach can identify common target genes in the development of resilience. Furthermore, studies in islets treated with ER stressors (chemical, lipids, cytokines, glucose, etc) when compared to islets from diabetes mouse models (such as Akita and NOD mice), can reveal similarities in the resilience programs following a systems biology approach of data analysis, including human islet data from healthy and diabetes subjects. We are happy that the reviewer agrees that this future direction is the ideal approach to take in order to assess the physiological relevance of our significant findings reported here.

In response to the reviewer's request, we determined some key elements that define β -cell resilience to ER stress in islets isolated from mice and treated with CPA.

We showed the following:

- (i) Treatment with CPA for 18h induced proinsulin accumulation in subcellular inclusions similar to those observed in MIN6 cells. (**Fig. 2G** of the revised manuscript and below).
- (ii) CPA caused a decrease in MAFA levels in β -cells, followed by quick recovery when CPA was washed out. (**Sup. Fig. 9A,B** of the revised manuscript and below).
- (iii) GSIS was blunted in CPA-treated cells and resumed recovery when CPA was washed out (**Sup. Fig. 9C**, in the revised manuscript and below).

We have included the following text in the manuscript describing Sup. Fig. 9 (page 25):

One limitation of our study is that while MIN6 cells provides an excellent system to discover factors involved in the resilience of β -cells to chronic ER stress, mimicking the range of stressors in human diabetes that include hyperglycemia, hyperlipidemia, cytokine action etc. or paracrine interactions in the islet environment is not possible (Brozzi and Eizirik, 2016). Future studies testing the effect of cytokines, lipids, glucose concentrations, etc. and can identify common target genes in the development of resilience. Furthermore, studies using mouse and human islets treated with ER stressors (chemical, lipids, cytokines, glucose, etc) and/or genetic models of stressed β -cells (Akita and NOD mice) have the potential to reveal similarities in the resilience programs following a systems biology approach of data analysis. Toward this direction, we showed that in mouse islets treated with CPA, levels of MAFA in β -cells decreased, followed by quick recovery when CPA was washed out. (**Sup. Fig. 9A,B**). Furthermore, GSIS was blunted in CPA-treated cells and resumed recovery when CPA was washed out (**Sup. Fig. 9C**).

Figure 2.

Supplementary Figure 9.

The changes to gene expression and cellular stress pathways can partly be addressed from the scRNAseq study where relative gene expression changes for alpha, delta and PP cells for T1D and T2D have been measured. Other than for beta cells the transcriptomes for the other endocrine cell types have scant mention. Are delta cell ER stress pathways equally affected?

We thank the reviewer for this question. In the original submission, we analyzed the expression of 46 genes of the ER protein processing pathway in T1D scRNA-seq β -cells (**Fig. 7A**). We agree with the reviewer that the analysis of additional islet cell types is important, therefore, in this revised manuscript, we extended our analysis of the 46 differentially expressed genes that we identified in β -cells, to α and δ -cells of healthy donors and T1D subjects (**Sup. Fig. 6A,B** of the revised manuscript and below).

We show that the relative expression of these genes in α and δ cells is different than β -cells. We made the following observations: (i) the expression of *BiP* was upregulated in all three cell types of T1D subjects, suggesting a chronic UPR response. (ii) all remaining genes except *BiP*, were downregulated, in β -cells in T1D subjects, suggesting decreased adaptation to ER stress (iii) a large number of genes were upregulated in α and δ cells, suggesting a partial sustained adaptation to ER stress. The function of the identified genes in the adaptation to ER stress in each islet cell type will need to be addressed in separate studies. However, our data conclude that β -cells in T1D subjects are likely more sensitive to ER stress, and lose their adaptive ER proteostasis mechanisms faster than other cell types during disease progression.

The following text has been added in the revised manuscript (page 17):

Strikingly, the relative expression of these genes in α -cells and δ -cells is different than β -cells. A large number of genes were upregulated in α and δ cells, among them *BiP*, suggesting a partial sustained adaptation to ER stress in these cell types (**Sup. Fig. 6A,B**). The function of the identified genes in the adaptation to ER stress in each islet cell type will need to be addressed in separate studies. However, our data conclude that β -cells in T1D subjects are likely more sensitive to ER stress and lose their adaptive ER proteostasis mechanisms faster than other cell types during disease progression.

Supplementary Figure 6.

Please give the source of the Min6 cells. A number of variant strains now exist in different labs with repeated passage and it is valuable to relate to other papers that used the same source.

We have provided the information in Materials and Methods. MIN6 cells were purchased from *AddexBio Technologies* (Catalog no.C0018008) and maintained according to the company's instructions. This information is added on page 31.

In Figure 1C a partial recovery of eIF2B is shown with washout after 12 h, but compared to control is it statistically any different at any time between 1 and 12h? If not then the recovery had occurred for all of these time points.

We evaluated the significance of changes in eIF2B activity between control and washout at 12h for statistical significance and no difference was found (**Fig. 1C** in the revised manuscript and below).

Figure 1.

Figure 11 – Is it possible to quantify the proinsulin puncta as in Fig. 2 since this would be valuable quantitative data?

We regret that we cannot provide quantification of the data in Fig. 11. Although the experimental conditions are identical between the two Figures, only the data in Fig. 2A-B were obtained with the confocal microscope. Data for Fig. 11 were obtained with a fluorescent microscope, which is not suitable for accurate quantification of the proinsulin inclusions. We kept Fig. 11 within the main Figures because we believe it nicely introduces the detailed analysis of Fig. 2A. However, we can move Fig. 11 to the supplemental materials if desired.

In response to the reviewer’s question, we decided to increase the number of cells analyzed in Fig. 2A. The differences in proinsulin localization in subcellular compartments became even more prominent when we used 5-fold more cells for the quantitative evaluation of subcellular localization. We have therefore replaced Fig. 2B with a new Figure, shown below.

Figure 2.

Reviewer #3 (Remarks to the Author):

The study by Chen et al relates to a relevant question for both type 1 and type 2 diabetes field, as ER stress is thought to be causative of beta-cell dysfunction in early developments of both conditions. However the findings reported are mostly corroborative of known mechanisms of beta-cell adaption to ER stress, and do not represent a significant advance as compared to the current knowledge in the field.

We apologize that we were not able to effectively communicate well the novelty of our findings in the original submission. As mentioned by reviewers 1 and 2, our study demonstrates the perseverance of β -cells to maintain function after exposure to near lethal ER stress conditions. Reviewer 2 suggested the term resilience which we think is very appropriate for the findings of this manuscript. There are no other studies on resilience of β -cells to ER stress. Our findings revealed signatures of loss of adaptation to ER stress in human islets and scRNA-seq analysis from T1D subjects. This is quite

remarkable that only β -cells showed the adaptive exhaustion phenotype. Reviewer 2 requested analysis of other cell types to show specificity of the loss of adaptation. Only β -cells had the severe inhibition of the adaptation program. These data are now included in the revised manuscript in **Sup. Fig. A,B**. Please see response to Reviewer 2.

We hypothesized that β -cells undergo cycles of stress and recovery in their normal life. We have simulated the cycles of stress and recovery in MIN6 cells by exposing cells to CPA and washing it out. Our data in MIN6 cells also suggest that when the stress recovery cycles fail, adaptation decreases and β -cell identity is not recoverable. Future studies might use a large cohort of healthy subject donors to determine heterogeneity of UPR gene expression in β -cells, and provide evidence for stress cycles in normal human islets.

In conclusion, our work brings innovative thinking to the field of stress resilience in β -cells. We have recently published another example of stress resilience in *Mol. Cell*, Oct 2021, under hyperosmotic stress. We want to share with the reviewer that we hope to review the literature along with the findings of our *Mol. Cell* paper and the *NC* paper under review, and write an article on *stress clocks, resilience and disease development*. We hope the reviewer likes our approach moving forward.

Major points:

1. *Lack of clear major novelty point or concept: it is already widely accepted that beta-cell adapt to ER stress by blocking mRNA translation initiation while simultaneously promoting the translation initiation of protective mechanisms (chaperones..). When this response is unable to resolve ER stress, the beta-cells become dysfunctional and/or die. There may also be an overuse of novel terminology (β EAR, transcriptional and translational reprogramming, beta-cell specific adaptosome, regulatome...) that makes it difficult for the reader to determine what is really relevant and novel.*

The reviewer correctly acknowledges the dogma in the field in translational control mechanisms during ER stress. This dogma is being repeated in every review written in the field. To our view, the adaptation to ER stress in the review articles and the current literature is oversimplified. In addition, the statement “unable to resolve stress” which is mentioned in all review articles, is not clear what it means. What the field has successfully done today, is to provide marker genes for the UPR. In addition, the field has provided data on the function of specific genes of the UPR in promoting or inhibiting adaptation by knocking them out or overexpressing them. As an example, the reviewer should notice the large number of papers that claim the transcription factor CHOP as pro-apoptotic. In contrast to this statement, we and in collaboration with Dr. Kaufman (PMID: 23624402) and others (PMID: 15601821) have shown that ATF4 and CHOP are promoting protein synthesis in cells under ER stress and do not bind promoters of pro-apoptotic genes. That’s why we called the high threshold of adaptation to ER stress in β -cells as self-defeating (PMID: 23645676). Furthermore, we published that PERK activation has unique features in inactivating eIF4E-dependent mRNA translation during

chronic ER stress, and reprogramming translation via the translation initiation factor eIF3 (PMID: 29220654). Therefore, the current model of the integrated stress response of translational inhibition and recovery via regulation of eIF2B activity (PMID: 29220654), is only accurate for acute ER stress, not chronic ER stress. Because chronic ER stress leads to disease development, we need to revise our assumptions of the mechanisms that operate during the chronic UPR.

We kindly ask the reviewer to reconsider the current literature reports. We provide new information looking at the regulation of groups of genes in specific cellular pathways during adaptation to ER stress. More importantly, we show the tremendous plasticity of the β -cells to undergo cycles of stress and recovery. In addition, we show unique features of proinsulin regulation during chronic ER stress at the level of transcription (decreased mRNA levels (Fig. 1G) with no change in proinsulin mRNA stability (data not shown)), translation, protein stability and subcellular localization. To our knowledge, these findings have not been reported before.

Overall, we hope that the reviewer can recognize that the simple view of the literature of the ER stress adaptation mechanisms is incompletely defined today. There are many factors, such as stress intensity, duration and coordinated pathway responses which need to be studied in detail to really give the correct answer to the question “**what is unresolved stress**”? This is the outstanding question that we have begun to address in this manuscript, thereby introducing the terms adaptive exhaustion and resilience to near lethal ER stress conditions.

2. The experimental model used (proliferative murine insulinoma cell line MIN6 subjected to a chemical treatment) is not very innovative as many studies have reported the effect of drug-induced ER stress on such model (MIN6 or INS1 with thapsigargin or tunicamycin treatment).

We apologize for not explaining clearly why we have used MIN6 cells. As we stated above, we tested mechanisms of resilience and adaptive exhaustion of β -cells in response to chronic ER stress. To answer this question, we need (i) a simple model system, such as the insulinoma MIN6 cells and (ii) a bona fide reversible ER stressor which promotes a synchronized response to stress. Furthermore, the ribosome profiling experiments we performed in MIN6 cells cannot be performed in islets. Even if we FACS-isolate human primary β -cells, it is not technically feasible to produce a pure population at the required scale necessary for our molecular analyses. We are confident that we did not repeat what others have shown, rather, we study the integrated signaling that leads to resilience and adaptive exhaustion.

Also the claims of ‘chronic’ vs ‘acute’ effects based on 18h and 2h treatment time is not justified. The authors should explain the choice for this drug/dose/timing.

We (for example, PMID: 29220654) and others (for example PMID: 12606582) have shown that the response to ER stress is temporal. We published a seminal paper describing for the first time the translational control mechanisms during the temporal response to stress. We have chosen the 1h and 18h time points based on previous publications by us and others (PMID: 29220654). Notably, the multiple cell types we assessed showed a near-identical pattern of temporal regulation of protein synthesis. As far as we know from our work and other publications, adaptation mechanisms to diverse stress conditions have a temporal response. The initial phase involves inhibition of protein synthesis and the adaptation phase involves translational recovery without the removal of the stress. This is why it is known as translational reprogramming in response to chronic stress. Among the different stress conditions that we showed temporal translational control are the stress of amino acid limitation and the stress of hypertonic stress (PMID: 34686314).

It's also crucial to describe the effect of the two CPA treatments on proliferation and cell death as these may influence the interpretation of some of the 'recovery' data presented throughout the manuscript. For instance, in Supp Fig 5C starting point is not shown, how much cell death there is post CPA6h, 18, and 42h treatment?

We thank the reviewer for this excellent question. In response to the reviewer's question, we plated equal number of cells and counted cells on the plates after CPA treatment. It is shown that treatment of MIN6 cells with CPA for 18h did not decrease the number of cells (**Sup. Fig. 7D** in the revised manuscript and below). In agreement with our previous publication (PMID: 29220654), longer treatments with CPA decreased cell viability (**Sup. Fig. 7C,D**). In addition, we did not observe any caspase 3 cleavage during treatment of MIN6 cells with CPA for 18h or longer times (of cells which remained attached on the plates) (**Sup. Fig. 7E** in the revised manuscript and below). Mouse Embryonic Fibroblasts treated with 600 mOsm hypertonic stress for 4h, known to induce apoptosis, were used as a positive control (PMID: 32175843). We emphasize that CPA treatment between 18-42h may be considered as a window of stress response reaching **unresolved stress conditions** that deserves further investigation. Our experimental system of 18h of CPA did not lead to conditions of unresolved stress and it therefore represents the window of adaptation to chronic ER stress.

Supplementary Figure 7.

It's also important to determine if beta-cell function (as assessed by glucose-stimulated insulin secretion) is altered in this model (or after repeated cycles of CPA treatment).

We thank the reviewer for this request. Additionally, we have addressed this question for reviewer 2. Please see response and **Sup. Fig. 1F and Sup. Fig. 9C**.

3. The strategy chosen of comparing CPA-treated MIN6 and MEFs is unclear and potentially misleading as the difference between the 2 lines is much more than only secretory vs non-secretory cells (primary vs transformed cells; embryonic vs adult; etc...). Also MEFs are often used as feeder cells given their capacity to secrete growth factors. Furthermore, conceptually it's not certain that the cause of ER stress-induced beta-cell failure are beta-cell specific genes.

We agree the reviewer has raised a logical concern. However, ER stress develops across different species and all cell types. Ideally, we would have liked to compare data from a collection of cell types to identify the unique features of the β -cell ER stress response. However, it is not practical to scale across many cell types, so we focused our comparison with MEFs. It is quite remarkable that we found a tremendous enrichment of ER-Golgi trafficking genes in MIN6 cells, which led us to further study the evolutionary conserved ER stress-induced proinsulin inclusions. We hope that the reviewer recognizes the inherent limitations of the detailed approaches employed in this study.

4. One can acknowledge the effort to relate the findings to human primary cells by using available datasets, however single-cell transcriptomics performed on human islet cells (that are not proliferative) subjected to the same CPA treatments would have been of great interest, in order to rule out transformed (proliferative) murine cell line-specific effects.

We thank the reviewer for the suggestion. We also performed experiments in mouse islets which we described in response to reviewer 2 (**Fig. 2G and Sup. Fig. 9A-C**). We plan to do the suggested studies in the future and compare with our current findings in scRNA-seq from human islets and MIN6 cells under CPA treatment.

5. The scRNAseq dataset of islets from T1D and non-diabetic individuals appears as one of the main novelty points of the study. However there are no information given on the characterization of the dataset (quality controls etc..) and the paper referred to (Kaestner et al 2019) is only a description of the HPAP consortium. It would be informative for the reader of this manuscript to know how the donors were selected as beta-cells were found in all 6 T1D donors included and at a rather normal frequency in most of the cases.

The HPAP consortium began accepting donors late 2016. At the time we accessed the dataset in 2020, we gathered all available scRNA-seq datasets that had been performed via identical methodology, which includes 6 T1D and 5 healthy donors as listed in **Sup. Fig. 5B** of the revised manuscript.

We first confirmed that there is no significant group variation within all scRNA-seq datasets. Then after quality control of the raw sequencing data, we filtered totally 35041 islet cells from 11 donors. The sequencing data were pooled and the normalization was done by SCtransform (Hafemeister and Satija 2019) in Seurat package in R. The normalized dataset contains 35041 islet cells, along with 23509 gene expression levels. The cell type was determined by Seurat clustering results and expression of pancreatic marker genes (PMID:27693023). To identify the specific cell types, we performed a non-supervised Uniform Manifold Approximation and Projection (UMAP; McInnes et al in 2018) of the dataset to cluster cells with similar expression profiles (see Figure below, labeled with INS gene as an example, for the reviewer only). We identified 8349 cells, which contain expression of known β -cell marker genes. These are the cells that we used to compare our data from MIN6 cells in this manuscript.

To understand the relationship between β -cell identity and UPR genes, we selected the expressions of β -cell markers, including INS, PAX6, NKX2.2, NKX6.1, MAFA and PDX1, with UPR genes (**GO:0030968**) of the 8349 β -cells and performed t-Distributed Stochastic Neighbor Embedding (tSNE) analysis by using Rtsne package in R. As shown in **Fig. 7**, there is a reverse correlation between β -cell identity gene expression and UPR genes.

We have updated the Material and Methods with more details as requested by the reviewer.

A number of additional analyses can be performed such as pseudo timing that may reveal different degrees of maturity of the beta-cells.

We performed pseudo timing analysis by in silico pseudo-time reconstruction in scRNA-seq analysis (TSCAN, by Zhicheng and Hongkai, 2016) and Monocle 3 (Cao et al, 2019) packages in R.

We calculated the existing tSNE data along with β -cell markers and UPR markers. We showed a gradual decrease of β -cell markers and a gradual increase of *BiP* in the pseudo timing analysis. We have included these data in **Fig. 7C**.

The following text was added in the revised manuscript (page 18):

Finally, in order to identify different degrees of maturity of β -cells in T1D subjects, we performed pseudo timing analysis by in silico pseudo-time reconstruction in scRNA-seq analysis (TSCAN, by Zhicheng and Hongkai, 2016) and Monocle 3 (Cao et al., 2019) packages in R. We calculated the existing tSNE data along with β -cell markers and UPR markers. We showed a gradual decrease of β -cell markers and a gradual increase of *BiP* in the pseudo timing analysis (**Fig.7C**).

Figure 7.

C.

On Fig 7B, there is no annotation of which cells belong to which donor, and which category of donors (ND, T1D).

We recognize the reviewer's concern about the strategy we used to analyze the scRNA-seq data. To make the presentation more clear, we collected clinical records of all 11 donors, re-analyzed the data and joined with the current de-dimensional tSNE map for a better presentation (**Sup. Fig. 6C** in the revised manuscript and below).

The following text was added in the revised manuscript (page17):

We also mapped individually the 11 donors (healthy and T1D), in the de-dimensional tSNE map for a better presentation. This analysis further supported the differential clustering of gene expression patterns for healthy and T1D subjects (**Sup. Fig.6C**).

Supplementary Figure 6.

6. The experimental set-up of ER stress cycles is potentially interesting but it's critical to know how much cell loss (death) there is per cycle. Some information on cell numbers (reflecting both proliferation and death) is needed. Along this line, cleaved Caspase 3 could be checked by WB at the different time points.

We agreed with the reviewer's comment that cell death may affect the results in the ER stress cycling experiment. To examine this, we performed the same experiment and quantified the number of attached cells and measured cleavage of Caspase 3.

We showed an increase of cell loss upon an increase of the stress cycles. However, no cleaved Caspase 3 was detected at the end of the 4th stress cycle, suggesting that the attached cells are resilient to cycles of ER stress. These data are shown in **Sup. Fig. 7F,G** in the revised manuscript and below.

The following text has been added in the revised manuscript (page 19):

We showed an increase of cell loss upon an increase of the stress cycles (**Sup. Fig.7F**). However, no cleaved Caspase 3 was detected at the end of the 4th stress cycle, suggesting that the attached cells are resilient to cycles of ER stress (**Sup. Fig. 7G**).

Supplementary Figure 7.

On Fig 7C, please show all data, also for cycles 2-5. How is the expression of the classical ER stress markers ATF3 and XBP1s/u (qPCR) and of insulin at protein level (WB)?

We include in the revised manuscript data on XBP1s and ATF3 (**Fig. 7E** of the revised manuscript and below). In our future studies we will comprehensively study the regulation of proinsulin/insulin in the stress cycles, including all levels of gene regulation (mRNA changes, translation, protein stability, subcellular localization). We thank the reviewer for the interesting comment.

Figure 7.

Other points:

1. One could expect that the loss of beta-cell maturity markers is associated with an increase in genes that are normally not expressed in adult beta-cells, such as progenitor markers (dedifferentiation) or an increase in markers specific for other islet cell types (transdifferentiation), as reported in a number of beta-cell dedifferentiation studies both in murine and human beta-cells (though the markers are different). Is that the case in these experimental models with (cycles of) CPA treatment?

We thank the reviewer for the comment. The temporal study of the progressive stress cycles is the continuation of this work. In the future, we plan to systematically perform the transcriptome/translatome identification during each stress cycle to identify gene expression patterns that correlate with diabetes development. This important follow up study was also suggested by reviewer 2.

2. Fig 1F: why is BIP already highly expressed in the control condition?

BiP is expressed in control condition because basal UPR in β -cells is required for their health (PMID: 34711668). Please notice, XBP1s protein is also present in control condition (**Sup. Fig. 7E** of the revised manuscript and above).

3. Fig 2: Claims made by the authors regarding the alteration in the processing of proinsulin (now done exclusively by fluorescence imaging) would be strengthened by a more standard biochemical approach (pulse-chase).

Please see our response to Reviewer 1. Data are shown in **Sup. Fig. 2**

4. Although the Discussion section is rather long, there is no discussion of the

limitations of the study.

We added the following paragraph at the end of the Discussion for the limitations of the Study (page 25):

One limitation of our study is that while MIN6 cells provides an excellent system to discover factors involved in the resilience of β -cells to chronic ER stress, mimicking the range of stressors in human diabetes that include hyperglycemia, hyperlipidemia, cytokine action etc. or paracrine interactions in the islet environment is not possible (Brozzi and Eizirik, 2016). Future studies testing the effect of cytokines, lipids, glucose concentrations, etc. and can identify common target genes in the development of resilience. Furthermore, studies using mouse and human islets treated with ER stressors (chemical, lipids, cytokines, glucose, etc) and/or genetic models of stressed β -cells (Akita and NOD mice) have the potential to reveal similarities in the resilience programs following a systems biology approach of data analysis. Toward this direction, we showed that in mouse islets treated with CPA, levels of MAFA in β -cells decreased, followed by quick recovery when CPA was washed out. (**Sup. Fig. 9A,B**). Furthermore, GSIS was blunted in CPA-treated cells and resumed recovery when CPA was washed out (**Sup. Fig. 9C**).

5. There is no quantification of the effect described on Fig 1I.

Please see our response to this question by Reviewer 2.

6. Supp Table 1 is missing.

We were unable to upload some files and Tables in the original submission. We will make sure all Tables and gene list tables are uploaded in the revised manuscript.

7. Gene lists corresponding to all main figures should be added as supplementary information.

Please see response to point 6.

8. Please justify how the 49 UPR-related genes have been defined.

The UPR gene reference was originally obtained from GO term: endoplasmic reticulum unfolded protein response (GO:0030968). It was previously used by other investigators, such as a recent publication in Science (PMID: 33384352).

REVIEWERS' COMMENTS

Reviewer #1 (Remarks to the Author):

The authors have well addressed all my comments. I have no further comments. Thanks for an interesting paper.

Reviewer #2 (Remarks to the Author):

The responses by the authors to the suggestions from the reviewers have been positive and additional data has been added. The revised manuscript is much improved.

Reviewer #3 (Remarks to the Author):

The authors answered the comments adequately and provided interesting additional data. No further comments.

Letter of Point-by-point Response to the Reviewers' Comments

REVIEWER COMMENTS

Reviewer #1 (Remarks to the Author):

The authors have well addressed all my comments. I have no further comments. Thanks for an interesting paper.

Reviewer #2 (Remarks to the Author):

The responses by the authors to the suggestions from the reviewers have been positive and additional data has been added. The revised manuscript is much improved.

Reviewer #3 (Remarks to the Author):

The authors answered the comments adequately and provided interesting additional data. No further comments.

We thank all reviewers whose comments and suggestions helped us to improve the quality of our manuscript.